# Accurate and scalable variant calling from single cell DNA sequencing data with ProSolo

David Lähnemann [1,2,3,4,5], Johannes Köster [5,6], Ute Fischer [4], Arndt Borkhardt [4], Alice C. McHardy [1,2,3,8 ✉] & Alexander Schönhuth [6,7,8 ✉]

Accurate single cell mutational profiles can reveal genomic cell-to-cell heterogeneity. However, sequencing libraries suitable for genotyping require whole genome amplification, which introduces allelic bias and copy errors. The resulting data violates assumptions of variant callers developed for bulk sequencing. Thus, only dedicated models accounting for amplification bias and errors can provide accurate calls. We present ProSolo for calling single nucleotide variants from multiple displacement amplified (MDA) single cell DNA sequencing data. ProSolo probabilistically models a single cell jointly with a bulk sequencing sample and integrates all relevant MDA biases in a site-specific and scalable—because computationally efficient—manner. This achieves a higher accuracy in calling and genotyping single nucleotide variants in single cells in comparison to state-of-the-art tools and supports imputation of insufficiently covered genotypes, when downstream tools cannot handle missing data. Moreover, ProSolo implements the first approach to control the false discovery rate reliably and flexibly. ProSolo is implemented in an extendable framework, with code and usage at: https://github.com/prosolo/prosolo

---

[1] Department for Computational Biology of Infection Research, Helmholtz Centre for Infection Research, 38124 Braunschweig, Germany. [2] Braunschweig Integrated Centre of Systems Biology (BRICS), Technische Universität Braunschweig, 38106 Braunschweig, Germany. [3] Algorithmic Bioinformatics, Faculty of Mathematics and Natural Sciences, Heinrich Heine University Düsseldorf, 40225 Düsseldorf, Germany. [4] Department of Paediatric Oncology, Haematology and Immunology, University Hospital, Medical Faculty, Heinrich Heine University Düsseldorf, 40225 Düsseldorf, Germany. [5] Algorithms for Reproducible Bioinformatics, Institute of Human Genetics, University of Duisburg-Essen, 45147 Essen, Germany. [6] Genome Data Science, Life Sciences Group, Centrum Wiskunde & Informatica, 1098 XG Amsterdam, The Netherlands. [7] Genome Data Science, Faculty of Technology, Bielefeld University, 33615 Bielefeld, Germany. [8] These authors jointly supervised this work: Alice C. McHardy and Alexander Schönhuth. ✉email: alice.mchardy@helmholtz-hzi.de; aschoen@cebitec.uni-bielefeld.de

Originally, genome sequences have been queried for genetic germline variation or for highly abundant somatic variation, for example in cancer. The advent of high-throughput single-cell sequencing has recently turned the spotlight on a type of so far understudied variation: the often less abundant somatic or post-zygotic variation that constantly accumulates with every mitotic cell division throughout the lifetime of an organism, turning every individual into a complicated genomic mosaic[1]. Estimates for somatic single nucleotide variants (SNVs) range from around $0.6 \times 10^{-9}$ up to $60 \times 10^{-9}$ mutations per genome position per cell division[2–5], with a recent estimate[6] based on single-cell sequencing at $2.66 \times 10^{-9}$. With the size of the human (reference) genome at $\sim 3.2 \times 10^9$ base pairs, these numbers indicate that even during healthy development, most cells harbor cell-specific point mutations. This enables retrospective monitoring of lineages involved in normal organism development, merely by sampling some cells[7], without having to interfere with its general development or having to kill the individual. In other words, this establishes a universally applicable methodology for in vivo lineage tracing.

In cancer development, this variation can be used to trace the cellular ancestry of tumor subclones and metastases[8–11], and to characterize the evolutionary dynamics of cancer progression[12,13]. In the long run, methods that account for the dynamics of mutational signatures in cellular evolution will improve the diagnosis, treatment, and prognosis of diseases for which somatic alterations are a key factor. To this end, obtaining accurate profiles of the genetic variation affecting single cells is essential.

In sequencing libraries prepared directly from single cells, only a small fraction of the genome is sampled. To obtain coverage levels that allow for the consistent identification of SNVs across larger parts of the genome, in vitro whole genome amplification is crucial. Among whole genome amplification methods, multiple displacement amplification (MDA[14]) has proven the least error-prone and is therefore considered the state-of-the-art in single-cell SNV profiling[15–18]. But, although the type of polymerase used in MDA ($\Phi$29) has the highest fidelity currently attainable (due to its proof-reading functionality), amplification errors still occur at a rate of $1.24 \times 10^{-6}$ to $9.5 \times 10^{-6}$ per copied base[15,19–22]—three orders of magnitude higher than the estimates for the somatic mutation rate. Further, the efficiency and fidelity of $\Phi$29 polymerase depends on the template sequence context[23], implying that the amplification error rate systematically varies around this average. Moreover, the degree of amplification depends on the quality of the template DNA extracted from the single cell[24] and how accessible each stretch of DNA is to amplification initiation via priming[25]. As a result, sequencing coverage after amplification differs both between sites along the genome and between the two alleles at a particular site, up to the entire dropout of alleles[26]. Because standard variant callers assume that alleles are uniformly covered, they do not perform well on the resulting data and are substantially outperformed by single-cell variant callers[27,28]. Clearly, when calling SNVs for single cells, the statistical uncertainties introduced by the amplification need to be dealt with at the largest possible accuracy.

Thus, variant callers for whole genome-amplified single-cell data need to account for both amplification errors and allelic bias, in addition to accounting for the site-specific variation. However, state-of-the-art single-cell SNV callers routinely assume fixed global rates when modeling uneven allelic coverage (up to dropout) and amplification errors. For example, to reflect the amplification error rate, both MonoVar[27] and SCcaller[28] work with global false-positive error rates for calling the presence of an alternative allele at a particular site. This assumes that $\Phi$29 polymerase is agnostic to local template sequence context,

although it is not[23]. Similarly, for modeling allele dropout, MonoVar[27] and SCIPhI[29] assume that one rate applies globally (and across all cells). This neglects that allele dropout, as the extreme case of uneven allele coverage, varies greatly along the genome and in particular also between cells, because their DNA is amplified separately. Interestingly, SCIPhI[29] additionally models allelic amplification bias to be governed by one global beta-binomial distribution (to apply for all cells in a dataset), thereby accounting for allelic dropout a second time (as the extreme values at 0 and 1 of that distribution). Two tools that more variably model allelic amplification bias are SCcaller[28] and SCAN-SNV[30]. Both estimate the minor allele frequency from nearby germline heterozygous sites. However, SCcaller employs a fixed global false-positive error rate for the calling of alternative alleles[28], and SCAN-SNV makes use of heuristics for filtering candidate variants[30]. Thus, to the best of our knowledge, there is no statistical model that allows for both local variation of bias and errors due to amplification, and for statistically sound false discovery rate control when calling and genotyping SNVs in single cells.

In this work we describe ProSolo, a variant caller using a unifying statistical framework that takes into account all relevant MDA-related biases and errors, allowing for them to vary locally. Importantly, our model enables a scalable—because computationally efficient—implementation, which is challenging even in bulk variant calling when considering local effects due to statistical uncertainties affecting the data[31]. ProSolo's statistical rigor allows for accurate control of the false discovery rate when calling alternative alleles or identifying other relevant effects, such as allele dropout. It achieves a higher variant calling accuracy compared to state-of-the-art tools.

## Results

**Comprehensive and flexible single-cell sequencing model.** We describe a probabilistic model that addresses the genotyping of diploid single cells whose DNA has been subject to whole genome multiple displacement amplification (MDA)[14]. Here, we name the central innovations of our model and demonstrate its advantages in comparison to existing approaches. A more detailed description of the innovations is available in the Methods section.

Briefly, our model addresses the two major issues of MDA: (i) the differential amplification of the two alleles present in a diploid cell ("amplification bias" in the following); (ii) MDA induced errors ("amplification errors" in the following) which are copy errors introduced by the $\Phi$29 polymerase used in MDA. To address amplification bias, we leverage a mechanistically motivated, empirically derived model of differential amplification of alleles. To assess amplification errors, we evaluate single-cell samples together with a bulk sample from which the single cell is supposed to stem. Regarding the latter, we argue that a bulk sample should be added to single-cell sequencing experiments wherever possible: it samples from the same cell population without requiring amplification, and is therefore unaffected by amplification bias and errors and thus makes a particularly useful background sample to address the statistical uncertainties and biases induced by MDA. At the same time, one of the major features of the core model and its implementation is that it can easily be adapted to flexibly deal with other sampling setups, so it could be extended to further scenarios.

**Consistently high alternative allele calling accuracy.** The most precise single-cell variant callers to date, SCcaller and SCIPhI, only call the presence vs. the absence of an alternative allele (i.e., the heterozygous and the homozygous alternative genotypes

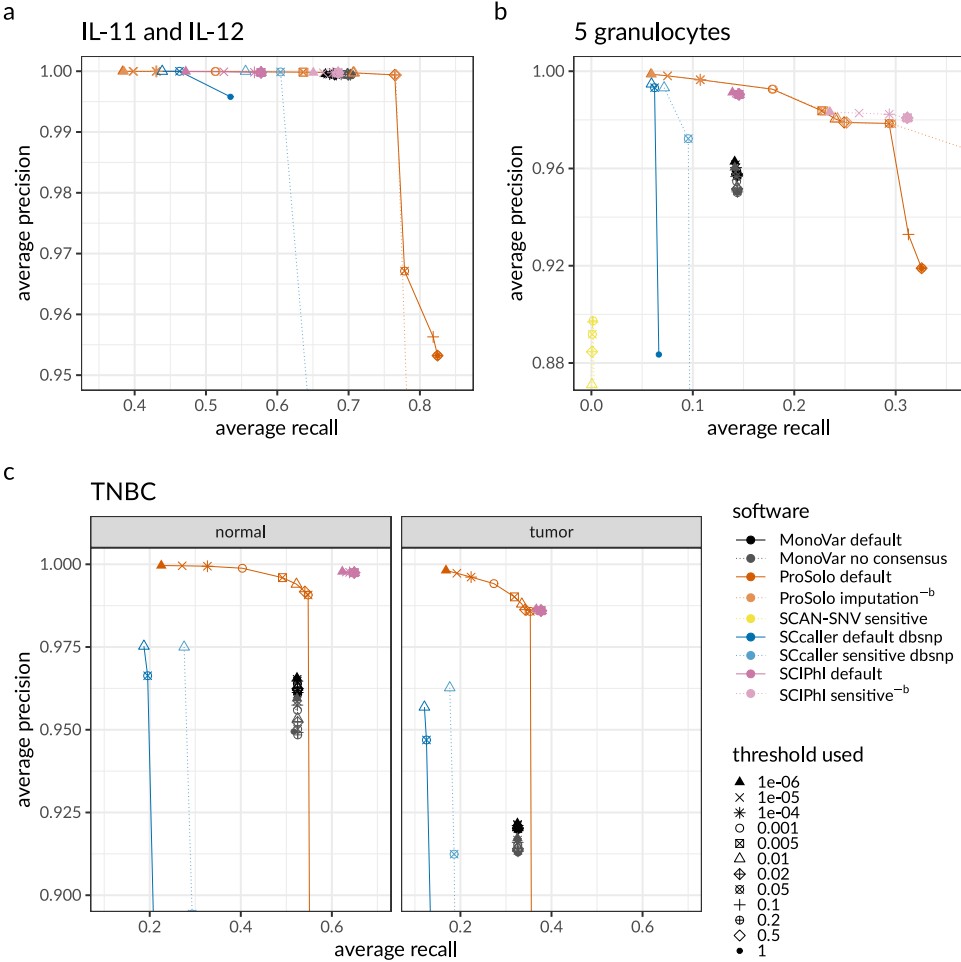

**Fig. 1 Precision-recall plots for alternative allele calls of ProSolo, MonoVar, SCAN-SNV, SCcaller, and SCIPhI.** All panels are strong zoom-ins, focusing on (different) areas of interest. Global views of these panels are provided in Supplementary Fig. 7. **a** Precision and recall of an average of two whole genome sequenced single cells IL-11 and IL-12 against their kindred clone IL-1C as ground truth genotypes. **b** Precision and recall average of the five whole-exome sequenced single granulocytes against their pedigree-based germline genotype ground truth. **c** Precision and recall average of 16 tumor and 16 normal single cells sequenced at the whole exome level. $^{-b}$ The germline ground truth induces an artificial increase of recall for SCIPhI's sensitive and ProSolo's imputation mode; these modes should thus be disregarded for a fair comparison on the granulocyte dataset in panel **b**. Threshold parameters (not comparable across tools): MonoVar --t; ProSolo --fdr; SCAN-SNV --fdr; SCcaller -a cutoff; SCIPhI prosolo --fdr. Software modes: MonoVar with consensus filtering (default) or without (no consensus); ProSolo with minimum coverage 1 in single-cell (default), or imputing zero coverage sites based on bulk sample (imputation); SCcaller with recommended settings (default) or with a more sensitive calling; SCIPhI with default parameters (default) or all heuristics off (sensitive).

called jointly). We thus focused on this for the main benchmarking.

The whole genome cell line dataset (see below, Methods section) seems much less challenging than the other dataset, as all methods achieved very high precision in alternative allele calling (Fig. 1a and Supplementary Fig. 7a), at recall rates of 0.45 and higher. In comparison to all other tools, ProSolo shows striking increases in the recall. For example, an increase of nearly 10% for precision above 0.99, where its maximum recall is 0.766, compared to 0.705 for MonoVar, 0.687 for SCIPhI, and 0.610 for SCcaller. The only exception with a recall of 0.0001 is SCAN-SNV (at a precision of 0.992). This can be explained by it aiming at somatic mutations, while the vast majority of SNVs in a genome will be germline variants.

Although a relative increase in recall of about 10% at utmost precision is certainly remarkable, ProSolo demonstrates its power on the second (whole exome) dataset (see below, Methods section). For this dataset, only SCcaller, SCIPhI, and ProSolo achieved a precision above 0.99, with ProSolo reaching a 20%

increase of recall to 0.178, compared to SCIPhI's 0.146, and SCcaller with 0.072 (Fig. 1b and Supplementary Fig. 7b). In comparison, MonoVar achieved a maximum precision of only 0.962. However, this was at a much higher recall (0.141) than for example SCcaller (0.095 at a precision of 0.972). SCcaller's decreased recall on this dataset might be due to its estimation of local allelic bias by also taking biases at neighboring sites into account—in whole-exome data the number of neighboring sites available for this estimation will be limited and might lead to less reliable estimates.

On this dataset, SCAN-SNV's recall increased to 0.0016 at a decreased maximum precision of 0.897. Most likely, this decreased precision is an artifact of using the germline genotype as ground truth. At the sites with somatic mutations in single cells, which SCAN-SNV focuses on, this ground truth will instead contain the homozygous reference germline genotype and will incorrectly classify (existing) alternative alleles as false positives. Due to this effect, we also expect the calculated precision of all the other tools to be underestimated. However, as the other tools also

provide alternative allele calls for all sites where the single cells retained this germline genotype, the relative effect on their precision will be smaller.

At the same time, this germline ground truth caveat also indicates that the recall of SCIPhI's sensitive mode and ProSolo's imputation mode will be an overestimate. Whenever coverage of a site is missing in a single cell, SCIPhI may impute the genotype with the last common ancestor genotype of the most closely related cells, while ProSolo will impute to the majority genotype in the bulk sample. Both strategies provide a biologically meaningful imputation that will be more useful than post hoc modes of imputation. However, at single-cell sites where a somatic mutation has created a true alternative allele, but no coverage is provided, we expect that both methods are most likely to call the homozygous reference germline genotype. In then comparing this to the germline genotype as the ground truth, these calls will be classified as true negatives even though they really constitute false negatives, thus artificially increasing recall. While the underestimation of precision equally affects all tools and generally means that benchmarking results are more conservative than with a more accurate (somatic) ground truth, this overestimation of recall in only these modes of two tools does not allow for a fair comparison. We have thus excluded both SCIPhI's sensitive mode and ProSolo's imputation mode from the discussion of the whole exome dataset with its germline ground truth (but their results are nevertheless displayed in Fig. 1b and Supplementary Figs. 7b, 9 for reference).

For the third dataset of single nucleus whole-exome sequencing data of 32 cells from a TNBC patient (16 tumor cells and 16 normal cells), we analysed tumor and normal cells separately (Fig. 1c). On the tumor cells, ProSolo is the only tool to achieve a precision above 0.99 (at a recall of 0.319). On the normal cells, both ProSolo and SCIPhI achieve a precision above 0.99. Here, SCIPhI outcompetes ProSolo with a maximum recall of 0.650 compared to ProSolo's 0.548. While ProSolo can achieve a similar recall (0.625, see –fdr threshold of 0.2 in Supplementary Fig. 7c), this comes at a reduced precision. This reflects the inherent uncertainty of the single-cell data and showcases a key difference in the approaches of ProSolo and SCIPhI. Where ProSolo models each cell and each genomic site separately, SCIPhI's model integrates information across all sites in all cells at once. While this can clearly help recover recall, this can also lead to false-positive somatic variant calls (Supplementary Fig. 13) and has clear implications for model complexity and software runtime. As all other tools, including ProSolo, are parallelizable over genomic regions and were thus able to process both datasets within days on a multicore machine, we do not report more detailed runtimes for them. However, it should be noted that SCIPhI took from 1 week (with iterations reduced below software defaults) on the single nucleus whole-exome dataset up to 7.5 weeks on the whole genome dataset, running on a single core without any possibility of parallelization. And where adding breadth of coverage (i.e., more genomic sites) or more cells simply means adding more parallel processes in most tools, both will further increase SCIPhI's wall time. In addition, adding more cells grows the space of possible tree topologies that SCIPhI explores super-exponentially[13], which will further increase its runtime.

**Flexible control over the false discovery rate**. Finally, a feature where ProSolo clearly stands out is the control over the false discovery rate. As can be seen in Fig. 1 (and Supplementary Fig. 7), ProSolo provides flexible control over precision vs. recall via specifying a false discovery rate of interest and is the only tool to achieve a precision of over 0.99 on the single tumor cell datasets from Wang et al.[21]. While no other tool provides a

formal control over the easily interpretable false discovery rate, several of the tools provide other types of thresholds that we varied in attempts to achieve higher precision or recall. However, none of them provide control over similar ranges of precision and recall. In addition, we ran the standard false discovery rate control implemented in ProSolo on the posterior probabilities provided by SCIPhI, but this did not provide any substantial control over the false discovery rate (Fig. 1 and Supplementary Fig. 14). The only limit to that range with ProSolo's current model is that it becomes less accurate when controlling for very small false discovery rates (below 0.01% for alternative allele calling in the whole genome dataset, see Supplement Supplementary Section 2.6). But this still leaves ProSolo as the only tool that provides the user with the choice of either aiming for more discoveries at the cost of a higher rate of false discoveries, or at aiming for a more limited number of discoveries with higher confidence in each of them.

**Valid model estimates of the allele dropout rate**. Leveraging our ground truths and using three different ways to calculate the allele dropout rate, we can confirm the general validity of our single-cell event definitions and also explore the limitations of the current model (Methods section). The expected allele dropout rates based on the ProSolo probabilities for allele dropout clearly fall into the range of previously published allele dropout rates[21,22,26,32–36] ("published" in Fig. 2). This analysis also clearly shows that the ProSolo expected allele dropout rates, based on the model's probabilities, correspond to those determined by comparing ProSolo genotypes with the ground truth (Fig. 2). This demonstrates that the explicit modeling of allele dropout events works and is useful for genotyping. However, in that comparison, the expected allele dropout rate was consistently underestimated on our own whole-exome data ("granulocytes", Fig. 2), and slightly overestimated for the data from Dong et al.[28] ("Dong 2017", Fig. 2). The comparison to a naively calculated allele dropout rate consistently shows an overestimation of the allele dropout probability, which is strongest for the samples with higher overall coverage (IL-11, PAG1, PAG10, Fig. 2 and Supplementary Fig. 3). However, this overestimation of the allele dropout probability does not seem to impact the genotyping resolution (Supplementary Figs. 15 and 16).

**Discussion**

ProSolo is the first method for SNV calling from MDA single-cell sequencing data to comprehensively model both amplification bias and amplification errors in a way that enables both the integration of site-specific differences (Fig. 3), and a computationally efficient—hence practically scalable—evaluation of all relevant data properties. We achieve this by combining a data-driven model of amplification errors (that incorporates a bulk background sample alongside the single-cell sample) with an empirical model of amplification bias which is dependent upon a site's coverage (and is based on a statistical understanding of the MDA process that is mechanistically motivated). The underlying model calculates posterior probabilities for fine-grained single-cell event definitions, whose false discovery rate can be controlled for and that can pass more information about data uncertainties to probabilistic models in downstream analyses; including methods that can then use this reduced, but highly accurate, data representation to integrate information across cells, for example for the phylogenetic reconstruction of cell relationships. Importantly, the calculation of posterior probabilities scales linearly in the coverage of variant sites, which is the fastest theoretically conceivable, documented by runtimes that have substantial advantages over the state-of-the-art in practice. Using one whole genome dataset

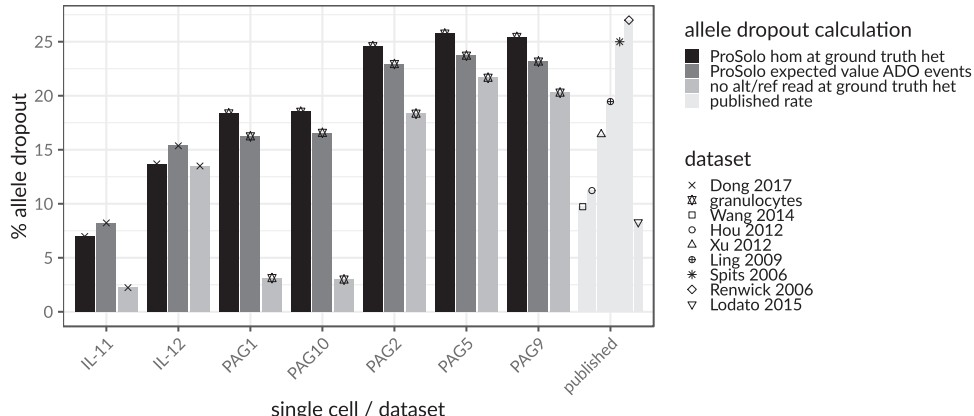

**Fig. 2 Concordance of three differently calculated per-cell (IL-1 and PAG cells) allele dropout rates across ground truth heterozygous sites, in the context of allele dropout rates from the literature[21,22,26,32–36].** The expected value of the allele dropout events in ProSolo (dark gray) is concordant with the number of false homozygous genotype calls made by ProSolo on those sites (black) and both values are well within the range of published allele dropout rates for single-cell MDA sequencing data ("published", to the right, very light gray). The naive allele dropout rate (light gray)—calculated as ground truth heterozygous sites with a minimum coverage of seven and either no read with the reference allele or no read with the alternative allele —-shows discrepancies with ProSolo's estimates of allele dropout for samples with a more uniform coverage (IL-11, PAG1, PAG10, Supplementary Fig. 3).

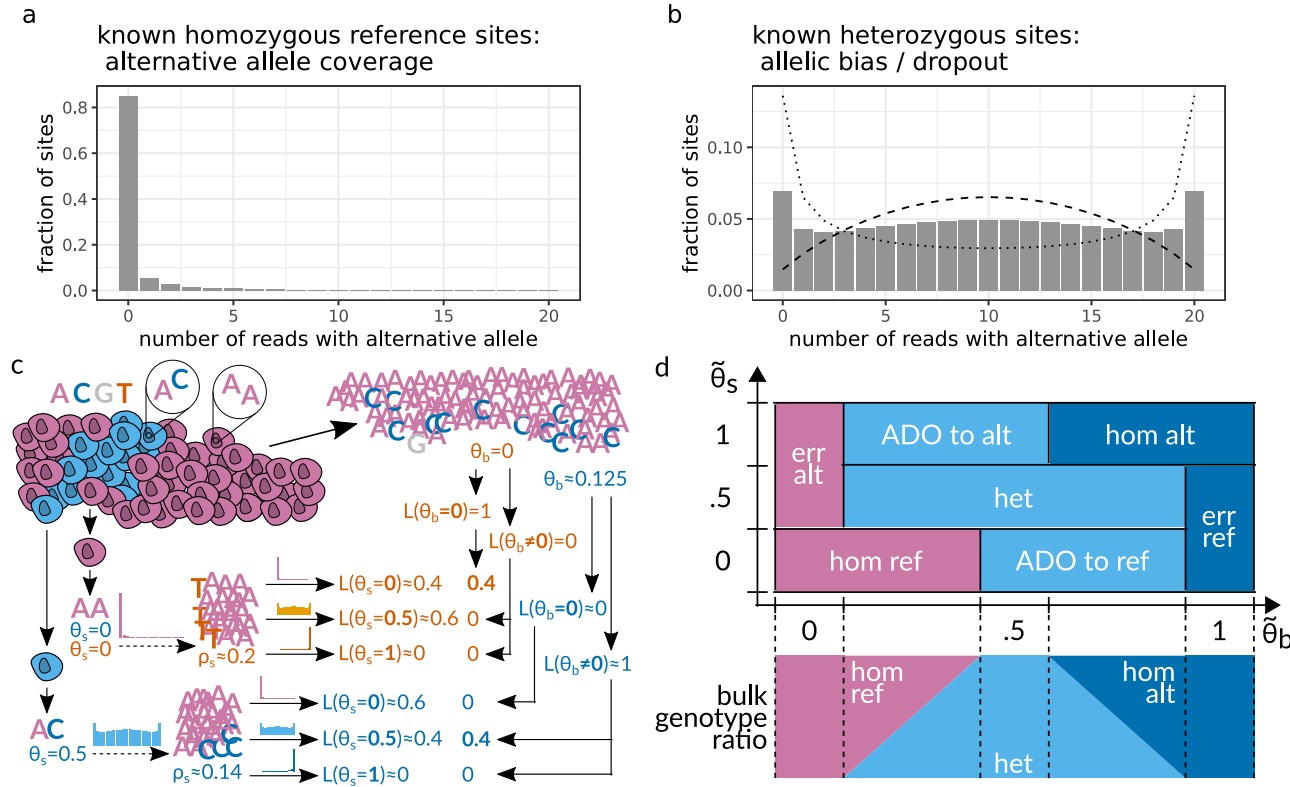

**Fig. 3 High-level overview of ProSolo's variant calling model. a, b** Exemplary alternative allele read count distributions for sites covered by 20 reads, as derived by Lodato et al.[22] Homozygous reference sites in **a** are assumed to follow a beta-binomial distribution; sites heterozygous for the alternative allele in **b** are assumed to follow the linear combination of two symmetrical beta-binomial distributions (dotted and dashed lines). **c** Toy example of calling the same genomic site in two single cells from the same population that differ in their true underlying allele frequencies for alternative allele C (blue, $\theta_s = 0$ vs. $\theta_s = 0.5$). Alternative nucleotide T (orange) is an amplification error. Empirical distributions in A and B account for the amplification bias, and likelihoods for the alternative allele candidates from the bulk reduce the likelihoods of amplification errors, thereby correctly identifying both the error and the original true mutation. This is formalized with the model in D. **d** Definition of single-cell events based on ProSolo's likelihood density estimates for the spectrum of true underlying alternative allele frequencies in the single-cell ($\tilde{\theta}_s$) and the bulk ($\tilde{\theta}_b$). The bulk is always assumed to be a combination of a maximum of two genotypes at a particular site, generating all possible $\theta_b$ (bottom panel). The model further assumes that the bulk sample has sufficient coverage to capture somatic variants. ADO allele dropout, alt alternative, err error, het heterozygous, hom homozygous, ref reference.

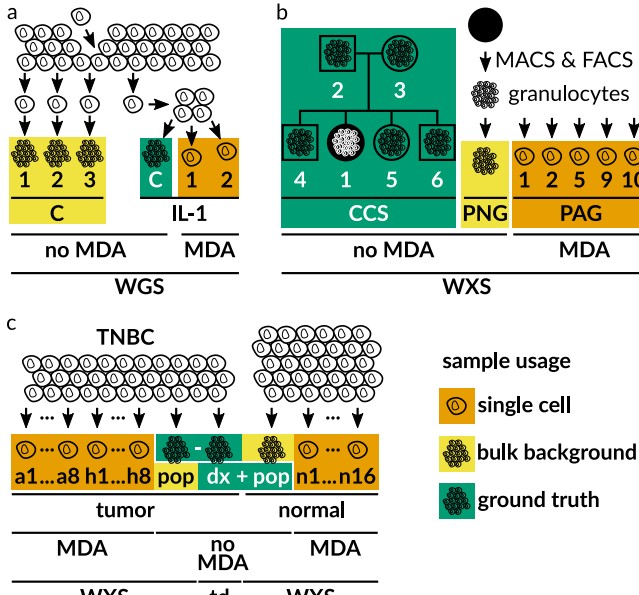

**Fig. 4 Benchmark Datasets.** For single cells, DNA was multiple displacement amplified (MDA). **a** Whole genome sequencing (WGS) dataset, generated from a clonal population started with an individual cell[28], and expanded further downstream to generate different bulk samples (C1, C2, C3, and IL-1C). **b** Newly generated whole exome sequencing (WXS) dataset of blood cells. Ground truth genotypes of patient CCS1 were determined from sequencing data of family members (boxes male; circles female). Granulocytes were isolated from blood using magnetic and fluorescence-activated cell sorting (MACS and FACS). **c** Whole exome sequencing (WXS) dataset from a triple-negative breast cancer (TNBC) patient[21]. For tumor cells (a1 to a8 and h1 to h8), the cell population (pop) sequencing of the normal sample was taken as the ground truth, including known clonal tumor-specific somatic variants previously validated by deep targeted duplex (td, dx) sequencing[21]. For normal cells (n1 to n16), the cell population (pop) sequencing of the tumor sample was taken as the ground truth, after removing known clonal and subclonal tumor-specific somatic variants previously validated by deep targeted duplex (td, dx) sequencing[21].

and two whole-exome datasets—each with a different type of ground truth (Fig. 4)—we demonstrate that these two innovations of ProSolo combine to achieve competing or better accuracy (Fig. 1). Moreover, the model allows to accurately control the false discovery rate of variant calls, another novelty of great practical value, and both the model and its modular implementation are easy to adapt and extend.

Especially the joint modeling of a single cell and a bulk sample is favorable from a statistical point of view: in terms of MDA induced bias and errors, the bulk acts as an unbiased sample of the population from which the single cell was drawn. Given enough bulk coverage (Supplementary Fig. 13), this provides a drastically more sensitive approach compared to, for example, the consensus rule of calling alternative alleles only when there is evidence in at least two (or even three) single cells (as implemented in MonoVar[27]). A more systematic and biologically relevant model of sharing information is implemented in SCIPhI, where the phylogenetic relationship of cells is part of the estimated parameters. Intuitively, if two cells are closely related, they have a higher likelihood of sharing a genotype at a particular site. However, both the consensus rule and the sharing of information via an inferred phylogeny requires that more than one single cell exhibit sufficient coverage of an alternative allele to call it reliably. This will rarely be achieved in single-cell MDA experiments, which are still often limited to a few dozen cells. In contrast, bulk

sequencing can easily be scaled to a much larger sampling of a cell population—representing hundreds or thousands of cells of a cell population—simply by adding a single higher coverage sample. Our benchmarking experiments demonstrated that ProSolo, by jointly modeling a single cell sample with a bulk sample, is the only tool to consistently achieve a precision above 0.99. In almost all datasets this precision is provided in concert with a substantially higher recall than any other tool. The only exception is the normal cells dataset from the single nucleus exome sequencing data, for which SCIPhI attains a higher recall above 0.99 precision—but at the cost of false-positive calls of subclonal heterozygous variants (Supplementary Fig. 13), while lacking the ability to call sites with a homozygous alternative genotype (Supplementary Fig. 13). Interestingly, a bulk coverage as low as 4X already provides drastic performance improvements in ProSolo's model and allows for a meaningful control over the false discovery rate (Supplementary Fig. 12; but higher bulk coverage will be necessary to detect low-frequency somatic variants, Supplementary Section 2.5). To summarize our benchmarking of alternative allele calling: SCcaller seems optimized for precision, MonoVar achieved higher recall than SCcaller, and SCAN-SNV does not seem suitable for general variant calling on MDA single-cell sequencing data but is only applicable when restricting interest to somatic variants. ProSolo clearly achieves the best precision. In two out of three datasets with fewer cells, this precision comes with a higher recall than in any other tool. This demonstrates, that ProSolo's model captures data from individual single cells better than any of the existing models. Only in the dataset with more cells and a bulk sample with lower coverage, SCIPhI can reach a higher recall than ProSolo by integrating information across cells (but at the cost of false-positive somatic mutation calls). However, with default configuration, SCIPhI takes weeks on a single core without any possibility of parallelization, compared to a runtime of only days on a cluster for all the other tools. This is the result of the general approach, integrating information across cells and across genomic sites all at once. Thus, any addition of genomic coverage and cells will further increase SCIPhI's wall clock runtime, where adding cells could prove especially troubling: adding cells will super-exponentially grow the space of possible tree topologies to explore. Other tools, including ProSolo, can simply add more parallel processes per site and cell and can thus leverage cluster computing for larger datasets, something that becomes more urgent with the advent of scalable droplet-based single-cell whole genome amplification methods[13].

Jointly calling single-cell variants with a bulk background sample also enables a biologically relevant imputation of genotypes at sites where single cells lack coverage. If genotype profiles without missing values are for example required by downstream tools, ProSolo can provide a cell population-specific imputation instead of resorting to a majority assignment (limited by the number of single cells sequenced) or even imputation based on external databases (e.g., dbsnp). However, any imputation will usually favor germline genotypes over low-frequency somatic genotypes (even though the opposite case also happens for SCIPhI, where some heterozygous subclonal variants are called in a lot more cells than the ground truth suggests, Supplementary Fig. 13). Thus, we suggest to not impute zero-coverage single cell sites whenever possible and instead recommend using and developing downstream software that can deal with both these missing values and the event probabilities that ProSolo provides. With this approach, uncertainties in the probabilities and information about missing data are passed on and will allow for more accurate statistical modeling in those downstream analyses.

Finally, ProSolo is the only method that provides a clearly interpretable false discovery rate parameter that can actually

control the trade-off between precision and recall (and this false discovery rate is the only parameter any user will have to specify). This can be used on alternative allele calls (Fig. 1, Supplement Supplementary Section 2.6), the main focus of our benchmarking, but also on any of the events pertinent to single-cell analysis (such as allele dropout or amplification errors) or combinations thereof (see Fig. 3d for all events). Thereby, beyond just calling alternative allele presence in a statistically reliable way, ProSolo can, for example, also compute the expected allele dropout rate across the entire genome of a particular cell in a robust manner (Supplementary Equation S 31 and Fig. 2).

ProSolo determines such global rates from its reliable site-specific posterior probabilities for all single-cell events that might be of interest, including allele-resolved genotypes, allele dropout, or amplification errors. We anticipate that such fine-grained site-specific probabilities—and the uncertainties they capture—will be informative for improving probabilistic modeling in downstream analyses. They could e.g., prove useful in models for phylogenetic reconstruction of the lineage relationship of sequenced single cells[29,37,38], thus achieving reconstruction results similar to or better than SCIPhI, while keeping the analyses modular and thus computationally tractable[13].

An in-depth look at one of the above-mentioned fine-grained events, allele dropout, showed that ProSolo's allele dropout rate estimates were within the range of published estimates. This confirms that our modeling of allele dropout events is realistic and can be useful in alternative allele calling and genotyping. However, the allele dropout rate estimations were slightly off in different directions for the different benchmarking datasets, suggesting that the empirical distributions we currently make use of (based on Lodato et al.[22]) may not suit all datasets. For example, the degree to which whole genome amplification introduces variability may be dataset-specific, something not currently captured in our model. This conclusion is further supported by the naively calculated allele dropout rate, which is much lower in the high coverage cells of the two datasets (IL-11 for the Dong et al. data[28], PAG1, and PAG10 in our granulocytes; see Supplementary Fig. 3). Interestingly, the sample IL-11 from Dong et al.[28] has been suggested to be a doublet[30], which might explain the higher overall coverage and points to a possible source for the discrepancy between the naively calculated and the estimated allele dropout rates. If samples PAG1 and PAG10 were doublets as well, this would indicate that our use of empirical distributions in ProSolo provides for more robust event probabilities in the presence of doublets, while heuristic thresholding (as in our naive allele dropout estimate) is very sensitive to such perturbations.

In general, the use of a fixed empirical model for MDA allelic bias does not seem to impede ProSolo's performance in alternative allele calling compared to the other tools, but has a noticeable effect on genotyping (Supplement, Supplementary Section 2.8, and Supplementary Fig. 13) and might be responsible for imprecisions when controlling for very small false discovery rates (Supplement, Supplementary Section 2.6). When future datasets are generated based on improved MDA protocols[13], these effects might be exacerbated. We, therefore, consider it important for future work to improve on the fitting of the empirical read count distributions observed when applying MDA to single cells (Supplement, Supplementary Section 1.2.2). For example, the parameters of the mechanistically motivated combination of beta-binomial distributions for modeling heterozygous genotypes could be learned per single cell sample at germline heterozygous sites—similar to the approaches of SCcaller and SCAN-SNV, but globally per cell with their local variation modeled by their dependence on a site's coverage.

In addition, the current model prevents the calling of subclonal somatic mutations with a bulk sample that has insufficient coverage to sample the respective allele, which can easily be remedied by sequencing the bulk sample to a greater depth. In addition, including a prior for the somatic mutation rate in the bulk sample further improve recall. Newer versions of the Varlociraptor library[31] already allow using such a prior, so this is immediate future work. When bulk coverage is low and an alternative allele is not sampled by any read, the prior will result in nonzero alternative allele frequencies in the bulk being assigned nonzero likelihoods, which is the desired behavior for this constellation. When bulk coverage is high, the increased amount of data will progressively overrule the prior.

Further room for improvement also remains for the modeling of the homozygous genotypes: while the current distributions account for both amplification and sequencing errors, sequencing errors are already safely accounted for elsewhere in our latent variable model. Beyond low coverage in the bulk background sample, this might be another reason why some subclonal somatic heterozygous variants are misclassified as homozygous by ProSolo (Supplement, Supplementary Fig. 13). Here, an amplification error profile that is not compounded with sequencing errors would need to be based on the $\Phi$29 polymerase error rate and a better understanding of the statistical distribution that these errors generate (Supplement, Supplementary Section 1.2.2). Studying and implementing the corresponding changes in the future has the potential to further improve the accurate site-specific event probabilities that ProSolo already provides through the joint modeling with a (sufficiently deep) bulk background sample.

Finally, the modular implementation of ProSolo within the context of the Varlociraptor library[31] facilitates the implementation of further features, such as read-backed phasing[39,40] or even more variable models of amplification bias[30] that could be integrated to exhaustively leverage MDA data information content. While no standardized software is available, it seems to be possible to obtain copy number profiles from single cells[41,42]. Extending our model to make use of such profiles, it should also be possible to salvage cancer-related variant cases that our current model setup does not capture (Supplement, Supplementary Section 1.5.1). And since the Varlociraptor library also provides advanced functionality for the calling of insertions, deletions and multiple nucleotide variants (MNVs), one of the immediately following next steps will be to adapt ProSolo to calling those in single cells.

Overall, ProSolo provides an accurate and easy-to-use variant caller for single-cell MDA sequencing data, which will easily scale to calling variation on more cells and broader genomic coverage. It will thus empower more research using single-cell DNA sequencing data.

## Methods

**Single-cell sequencing model**. Here, we introduce the central innovations of our probabilistic model for genotyping of diploid single cells whose DNA has been subject to whole genome multiple displacement amplification (MDA)[14]. More details and a detailed derivation of all model elements can be found in the Supplement, including a summary of the core model.

We address amplification bias by leveraging a mechanistically motivated, empirically derived model of differential amplification of alleles and assess amplification errors by evaluating single-cell samples together with a bulk sample from which the single cell is supposed to stem. Regarding the latter, we argue that a bulk sample should be added to single-cell sequencing experiments wherever possible: it samples from the same cell population without requiring amplification, and is therefore unaffected by amplification bias and errors and thus makes a particularly useful background sample to address the statistical uncertainties and biases induced by MDA. For related work on flexible bulk sequencing sample composition, see Köster et al.[31].

**Table 1 Parameters of the multiple displacement amplification (MDA) amplification bias model as presented in Supplementary Fig. S5, panels C and D of ref. [22].**

| parameter | slope | intersect |
|---|---|---|
| $\alpha$ | −0.00003 | 0.06857 |
| $\beta$ | 0.00745 | 2.36749 |
| $w$ | 0.00055 | 0.54040 |
| $\alpha_1$ | 0.05738 | 0.66973 |
| $\alpha_2$ | 0.00323 | 0.39926 |

The full precision as used in the implementation is given in Supplementary Supplementary Table 1.

*A mechanistically motivated model of amplification bias, trained on data, gives realistic coverage-specific single-cell (genotype) likelihoods.* To account for MDA amplification bias up to the complete dropout of individual alleles, we distinguish between two alternative allele frequencies:

(i) The true (but usually unknown) underlying allele frequency at a site in a single cell: $\theta_s$. This can be assigned one of three possible values, namely $\theta_s \in \{0, 0.5, 1\}$, where 0 and 1 represent the homozygous reference and alternative genotype and 0.5 a heterozygous genotype. However, the ratio of reads harboring the different alleles from a single cell sequencing experiment does not reflect the true allele frequency, because of the biases induced by amplification. Instead, the ratio of reads reflects

(ii) the allele frequency after its distortion through amplification bias. For a site with total coverage of $l$ reads, of which $k$ reads bear the alternative allele, the formal definition of this measurable frequency is $\rho_s = \frac{k}{l}$.

The goal is to estimate the likelihood density across the three possible underlying allele frequencies in the single-cell (we denote $\bar{\theta}_s$ as the density estimate across all $\theta_s \in \{0.0, 0.5, 1.0\}$). To accurately quantify the uncertainty introduced by amplification bias, we consider

$$\mathbf{P}(\rho_s|\theta_s) \qquad \text{for} \quad \theta_s \in \{0, 0.5, 1\}, \qquad (1)$$

the probability distributions that reflect the shift from the true allele frequency $\theta_s$ to the distorted allele frequency $\rho_s$, as induced by MDA (Fig. 3c). We thus formally describe the statistics of read counts skewed by MDA at all sites, encompassing sites that are homozygous for the reference allele, heterozygous, or homozygous for the alternative allele, to calculate likelihoods for each of these possible true allele frequencies.

To do this, we follow the considerations of Lodato et al.[22] (see Fig. S5 and the section "Modeling MDA-derived alternative read counts" of the respective supplement for the details), who fitted well-studied probability distributions to empirical distributions they obtained for $\mathbf{P}(\rho_s|\theta_s)$ of $\theta_s \in \{0, 0.5\}$. For sites that are homozygous for the reference allele, amplification bias cannot initially happen. However, once an amplification error creates an alternative allele, this can be amplified to large frequencies due to amplification bias. Lodato et al.[22] thus consider the empirical distribution $\mathbf{P}(\rho_s|\theta_s = 0)$ to follow a beta-binomial. Effectively, this means that the probability of a nonzero alternative read count ($\rho_s > 0$, which is an alternative read count above 0 in Fig. 3a) will be nonzero ($\mathbf{P}(\rho_s|\theta_s = 0) > 0$), merely because of sequencing and amplification errors. In contrast, at heterozygous sites (Fig. 3b), the distribution is dominated by amplification bias. Thus, for $\mathbf{P}(\rho_s|\theta_s = 0.5)$ they found a mixture of two beta-binomial distributions to appropriately fit the empirically observed distributions (Fig. 3b). We further motivate the choice of the beta-binomial distribution mechanistically by an analogy to its generative urn model named after Pólya[43]: Take an urn with two white and two black balls, where each ball represents one strand of each (double-stranded) allele at a heterozygous site. In the Pólya urn model, drawing a ball leads to the replacement of that ball with two balls of the same color, analogous to a strand copy by the Φ29 polymerase (additional discussion in the Supplement, Supplementary Section 1.2.1). Finally, the necessity for a mixture of beta-binomials (Fig. 3b) becomes evident when contrasting it with the binomial distribution observed at heterozygous sites in bulk experiments. Namely, the bulk distribution for reads supporting the alternative allele would narrowly peak at a count of $k = \frac{l}{2}$, which is not the case for the corresponding $k = 10$ in Fig. 3b. Instead, the mixture of beta-binomials peaks at the extreme read counts of $k = 0$ and $k = l$, highlighting that the dropout of the alternative or reference allele is quite likely to occur. Most likely, this mixture arises from differences in the accessibility of different pieces of the genomic DNA for amplification[25] (Supplement, Supplementary Section 1.2.1). Finally, for homozygous alternative sites (i.e., $\mathbf{P}(\rho_s|\theta_s = 1)$), we rely on the symmetry of the

cases $\theta_s = 0$ and $\theta_s = 1$. In summary, we obtain the following equations for $\mathbf{P}(\rho_s|\theta_s)$:

$$
\begin{aligned}
\mathbf{P}\left(\rho_s = \frac{k}{l}\middle|\theta_s = 0\right) &= \mathbf{BB}(k, l; \alpha(l), \beta(l)), \\
\mathbf{P}\left(\rho_s = \frac{k}{l}\middle|\theta_s = \frac{1}{2}\right) &= w(l) \times \mathbf{BB}(k, l; \alpha_1(l), \alpha_1(l)) \\
&\quad + (1 - w(l)) \times \mathbf{BB}(k, l; \alpha_2(l), \alpha_2(l)), \\
\mathbf{P}\left(\rho_s = \frac{k}{l}\middle|\theta_s = 1\right) &= \mathbf{BB}(k, l; \beta(l), \alpha(l)).
\end{aligned}
\qquad (2)
$$

Here, $\mathbf{BB}$ represents the beta-binomial probability mass function where $k \in \{0, ..., l\}$ with $l$ being the total read coverage of the site that is considered. All of parameters $\alpha$, $\alpha_1$, $\alpha_2$, $\beta$, and $w$ scale linearly in $l$. Allowing to vary distributions through these parameters allows amplification bias to depend on the total coverage of a site, and thereby to vary locally. The symmetry of distributions for the homozygous sites ($\theta_s = 0$ in Fig. 3a, and $\theta_s = 1$) is established by swapping the shape parameters $\alpha(l)$ and $\beta(l)$. The distribution for heterozygous sites (Fig. 3b) corresponds to a mixture of two beta-binomials with shape parameters $\alpha_1(l) = \beta_2(l)$ and $\alpha_2(l) = \beta_2(l)$, where equality of $\alpha(l)$ and $\beta(l)$ yields symmetry of the distributions relative to $k = \frac{l}{2}$ (the expected value for alternative read counts at heterozygous sites). Slopes and intersects for the scaling of all parameter values across different choices of $l$ are given in Table 1.

*Using bulk evidence of alternative alleles to increase the accuracy of variant calls.* A bulk sequencing sample of the same cell type from the same organism is a much larger sampling of that cell population than the sequencing of dozens of single cells. Unless a single cell is the only one to harbor a particular mutation, a deep enough sequencing of the bulk sample from which the single cell was drawn should contain reads from cells that share the particular mutation with the single cell. Thus, considering bulk samples corresponds to drawing unbiased samples of the entire population (of genome copies), in contrast to single cells that correspond to heavily distorted measurements on pairs of copies. This establishes a formal, statistical argument why one should consider bulk experiments in single-cell sequencing whenever possible: the identification of a mutation in an accompanying bulk sample lends further credibility to the mutation in the single cell, in a data-driven way, without assuming any fixed error rate (Fig. 3c). At the same time, the absence of a single cell candidate mutation in a bulk sample (with sufficient sequencing depth) increases the probability of an amplification error. As a consequence, bulk samples can be employed to improve both the sensitivity and specificity of variant calls in the single cell, where the increasing depth of coverage of the bulk sample increases the accuracy of the calls.

For our model, we derive likelihood density estimates for all possible alternative allele frequencies in the background bulk sample. Given a set of $n$ reads from the bulk ($b$) read data $\mathbf{Z}^b = \{\mathbf{Z}_1^b, \mathbf{Z}_2^b, \dots, \mathbf{Z}_n^b\}$, and discrete possible allele frequencies $\frac{m}{n}$ ($m \in 0, 1, \dots, n$), we compute the probability of the data given a particular allele frequency as the product of the probabilities of all the reads:

$$L\left(\theta_b = \frac{m}{n}\middle|\mathbf{Z}^b\right) \propto P\left(\mathbf{Z}^b\middle|\theta_b = \frac{m}{n}\right) = \prod_{j=1}^{n} P\left(Z_j^b\middle|\theta_b = \frac{m}{n}\right) \qquad (3)$$

Here, the probability of an individual read, given a particular allele frequency, $\mathbf{P}(Z_j^b|\theta_b = \frac{m}{n})$, is defined as in Supplementary Equation S 21, based on the model described in Köster et al.[31]. Further, as for the single-cell sample, when referring to the likelihood density estimates across all possible allele frequencies (as opposed to a particular allele frequency), we denote this with $\bar{\theta}_b$.

*Calculating posterior probabilities for events at single-cell sites, including a bulk background sample.* With single-cell genotype likelihoods adjusted by our empirical amplification bias model (Eq. (2)) and auxiliary evidence on alternative alleles from a bulk sample (Eq. (3)), we define mutually exclusive single-cell events. Figure 3c gives a simplified illustration of how the combination of likelihood density estimates across all possible allele frequencies $\bar{\theta}_s$ and $\bar{\theta}_b$ works for calling a heterozygous and a homozygous genotype in a single cell. However, our model fully defines the two-dimensional space of possible underlying alternative allele frequencies in the two samples as:

$$E = \left\{0, \frac{1}{2}, 1\right\}_{\bar{\theta}_s} \times [0, 1]_{\bar{\theta}_b}. \qquad (4)$$

For these definitions, we always assume that—regarding a particular genomic site—the bulk cell population can only consist of a maximum of two subpopulations that are exactly one mutated allele apart from each other (Fig. 3d, lower panel; for discussion on the validity of this assumption, see Supplementary Section 1.5.1). At allele frequencies $\theta_b \in \{0, 0.5, 1\}$, we assume a homogeneous population of only homozygous reference, heterozygous, or homozygous alternative cells, respectively. At frequencies in between, we assume a mixture of heterozygous cells with homozygous reference cells ($\theta_b \in (0, 0.5)$), or with homozygous alternative cells ($\theta_b \in (0.5, 1)$). Bearing this assumption in mind, we define single-cell events as mutually exclusive two-dimensional sub-spaces. For

example, an error-free homozygous alternative site is defined by allele frequency likelihood density estimates across $E_{\text{hom alt}} = \{1\}_{\hat{\theta}_s} \times (\frac{1}{2}, 1]_{\hat{\theta}_b}$. And the dropout of an alternative allele is defined across allele frequency likelihood density estimates $E_{\text{ADO to ref}} = \{0\}_{\hat{\theta}_s} \times [\frac{1}{2}, 1)_{\hat{\theta}_b}$ (Fig. 3d and Supplementary Table 2). For the latter, please note that we here resolve a contradiction between $\tilde{\theta}_s = 0$, which indicates a homozygous reference single cell, and $\hat{\theta}_b \in [0.5, 1)$, which indicates that the bulk does not contain any homozygous reference cells. We decide to trust the bulk sample over the single-cell sample, and with our above assumption that the bulk is a mixture of a maximum of two subpopulations, the bulk can only contain heterozygous and homozygous alternative cells. This renders a homozygous reference single-cell impossible, and we classify this event as evidence for an allele dropout of the alternative allele.

We thus obtain a set of mutually exclusive single-cell events (Fig. 3d):

$$\mathfrak{E} = \{E_{\text{hom ref}}, E_{\text{err alt}}, E_{\text{ADO to alt}}, E_{\text{het}}, E_{\text{ADO to ref}}, E_{\text{err ref}}, E_{\text{hom alt}}\} \tag{5}$$

Assuming a flat prior across the possible underlying allele frequencies for both the bulk and the single cell, we can compute likelihoods for all those single-cell events (e.g., Supplementary Equation S 28). The sum of the likelihoods of all these (mutually exclusive) events yields the marginal probability (Supplementary Equation S 26). Using the marginal probability, we can calculate the posterior probability for any of these events. For example, the posterior probability of event $E_{\text{hom alt}}$ (Fig. 3d) can be calculated with:

$$\begin{aligned}
\mathbf{P}(E_{\text{hom alt}}|\boldsymbol{Z}^s, \boldsymbol{Z}^b) &= \frac{1}{\sum_{E_e \in \mathfrak{E}} \mathbf{P}(\boldsymbol{Z}^s, \boldsymbol{Z}^b|E_e)} \times \mathbf{P}(\boldsymbol{Z}^s, \boldsymbol{Z}^b|E_{\text{hom alt}}) \\
&= \frac{1}{\sum_{E_e \in \mathfrak{E}} \mathbf{P}(\boldsymbol{Z}^s, \boldsymbol{Z}^b|E_e)} \times \mathbf{P}(\boldsymbol{Z}^s|\theta_s = 1) \times \mathbf{P}(\boldsymbol{Z}^b|\theta_b \in (\tfrac{1}{2}, 1])
\end{aligned} \tag{6}$$

Accounting for the sample likelihoods based on Supplementary Equation S 23 (assuming $\rho_b = \theta_b$ for the bulk that has no amplification step, Supplementary Equation S 3), and evaluating only point estimates of these likelihoods at possible alternative allele frequencies, this gives:

$$\begin{aligned}
\mathbf{P}(E_{\text{hom alt}}|\boldsymbol{Z}^s, \boldsymbol{Z}^b) &= \frac{1}{\sum_{E_e \in \mathfrak{E}} \mathbf{P}(\boldsymbol{Z}^s, \boldsymbol{Z}^b|E_e)} \\
&\times \sum_{k=0}^{l} \left\{ \mathbf{P}(\boldsymbol{Z}^s|\rho_s = \tfrac{k}{l}) \times \mathbf{P}(\rho_s = \tfrac{k}{l}|\theta_s = 1) \right\} \\
&\times \sum_{m=\frac{n}{2}}^{n} \mathbf{P}(\boldsymbol{Z}^b|\theta_b = \tfrac{m}{n})
\end{aligned} \tag{7}$$

Accounting for amplification bias with Eq. (2) and computing the likelihood of the sample-specific allele frequency ranges with Supplementary Equation S 22 and Eq. (3), this becomes:

$$\begin{aligned}
\mathbf{P}(E_{\text{hom alt}}|\boldsymbol{Z}^s, \boldsymbol{Z}^b) &= \frac{1}{\sum_{E_e \in \mathfrak{E}} \mathbf{P}(\boldsymbol{Z}^s, \boldsymbol{Z}^b|E_e)} \\
&\times \sum_{k=0}^{l} \left\{ \prod_{i=1}^{l} \mathbf{P}(Z_i^s|\rho_s = \tfrac{k}{l}) \times \mathbf{BB}(k, l; \beta(l), \alpha(l)) \right\} \\
&\times \sum_{m=\frac{n}{2}}^{n} \prod_{j=1}^{l} \mathbf{P}(Z_j^b|\theta_b = \tfrac{m}{n})
\end{aligned} \tag{8}$$

With read probabilities calculated with Supplementary Equation S 21, we can calculate the posterior probability of the $E_{\text{hom alt}}$ event (for an analogous and more detailed derivation for $E_{\text{ADO to ref}}$, see Supplement, Supplementary Section 1.4). To obtain posterior probabilities for compound events, we sum up the posterior probabilities of the events it comprises, effectively joining up their respective allele frequency range combinations into compound range combinations. For the site-specific single-cell probability for the presence of an alternative allele in the single cell, we thus get the compound event (blue events in Fig. 3d and Supplementary Table 2):

$$\begin{aligned}
E_{\text{alt presence}_s} &= E_{\text{ADO to ref}} \cup E_{\text{err ref}} \cup E_{\text{het}} \cup E_{\text{ADO to alt}} \cup E_{\text{hom alt}} \\
&= \{0\}_{\theta_s} \times \left[\tfrac{1}{2}, 1\right)_{\theta_b} \cup \left\{0, \tfrac{1}{2}\right\}_{\theta_s} \times [1]_{\theta_b} \cup \left\{\tfrac{1}{2}\right\}_{\theta_s} \times (0, 1)_{\theta_b} \cup \\
&\quad \{1\}_{\theta_s} \times \left(0, \tfrac{1}{2}\right]_{\theta_b} \cup \{1\}_{\theta_s} \times \left(\tfrac{1}{2}, 1\right]_{\theta_b} \\
&= \{0\}_{\theta_s} \times \left[\tfrac{1}{2}, 1\right]_{\theta_b} \cup \left\{\tfrac{1}{2}, 1\right\}_{\theta_s} \times (0, 1]_{\theta_b}
\end{aligned} \tag{9}$$

whose posterior probability we can obtain from this sum:

$$\begin{aligned}
\mathbf{P}(\text{alt presence}_s|\boldsymbol{Z}^s, \boldsymbol{Z}^b) &= \mathbf{P}(E_{\text{ADO to ref}}|\boldsymbol{Z}^s, \boldsymbol{Z}^b) + \mathbf{P}(E_{\text{err ref}}|\boldsymbol{Z}^s, \boldsymbol{Z}^b) + \mathbf{P}(E_{\text{het}}|\boldsymbol{Z}^s, \boldsymbol{Z}^b) \\
&\quad + \mathbf{P}(E_{\text{ADO to alt}}|\boldsymbol{Z}^s, \boldsymbol{Z}^b) + \mathbf{P}(E_{\text{hom alt}}|\boldsymbol{Z}^s, \boldsymbol{Z}^b)
\end{aligned} \tag{10}$$

To genotype, we calculate the posterior probability of all three possible single-cell genotypes and choose the genotype with the maximum posterior probability. In Fig. 3d, events are colored purple when implying the homozygous reference genotype, light blue when implying the heterozygous genotype, and dark blue when implying the homozygous alternative genotype (Supplementary Table 2). Similarly, to calculate the posterior probability of an allele dropout at a particular site, we sum up the posterior probabilities of the two ADO events.

For any such compound event, we can estimate a threshold on the posterior probabilities that controls for a specified false discovery rate. This is based on the approach described by Müller et al.[44]—for further details see the Supplement (Sections S 1.6 and Supplementary Fig. 2.6) and Köster et al.[31].

*Biologically relevant imputation based on the bulk sample.* As argued above, a large enough sampling of the bulk cell population that the single-cell comes from should contain the single cell's genotype at a particular site, unless this cell is genuinely the first cell to harbor a mutation at that site. This bulk background sample can thereby render credibility to single-cell variants with low coverage, while at the same time eliminating amplification errors in the single-cell sample, as these will not exist in the bulk sample. Interestingly, the bulk sample also provides a mechanism of biologically meaningful imputation at sites that have no read coverage in the single cell. If imputation is desired for sites with no read coverage in the single cell, we set $\mathbf{P}(\boldsymbol{Z}^s|\theta_s = 0) = \mathbf{P}(\boldsymbol{Z}^s|\theta_s = \tfrac{1}{2}) = \mathbf{P}(\boldsymbol{Z}^s|\theta_s = 1) = 1$, rendering all (unknown underlying) single-cell genotypes equally likely. Thus, the posterior probabilities of events at sites with no read coverage become solely dependent on the read data from the bulk sample, providing the most common genotype in the bulk. However, while this is a biologically meaningful way of imputation at the vast majority of genomic sites, it should be noted that this imputation will favor germline genotypes over any existing (lower frequency) somatic genotypes at a site, unless such a somatic genotype is present in a majority of cells. Thus, such an imputation carries the potential to introduce erroneous calls (especially when looking at subclonal somatic mutations), and we recommend to instead use downstream tools that can accommodate for missing data and data uncertainty wherever they are available.

*ProSolo is an easy-to-use command-line tool, based on a modular framework.* ProSolo is an easy-to-use command-line tool—following usability standards[45]—and its source code is available at https://github.com/prosolo/prosolo, including instructions for an easy installation via Bioconda[46]. Its main contribution in terms of software is the implementation of its comprehensive statistical model into the Rust variant calling library of Varlociraptor[31]. See Supplement (Supplementary Section 1.6) for further implementation details.

**Benchmarking.** We compare ProSolo to the state-of-the-art for SNV calling from single-cell sequencing data of multiple displacements amplified (MDA) DNA: MonoVar[27], SCAN-SNV[30], SCcaller[28], and SCIPhI[29]. We used Snakemake[47] (version 6.3.0) to implement the benchmarking workflows. For detailed information on benchmarking setup and results, see Supplementary Supplementary Section 2. All code used for benchmarking is available at: https://github.com/prosolo/benchmarking_prosolo(or as preserved by Zenodo at: https://doi.org/10.5281/zenodo.3769115).

*Datasets and ground truths.* We benchmarked ProSolo on three experimental datasets (Fig. 4), each with a different kind of ground truth:

Whole genome sequencing of almost identical kin-cells from a cell line[28]. The first dataset comes from the publication of the SCcaller software[28] (Fig. 4a, dataset available from SRA project PRJNA305211 [https://www.ncbi.nlm.nih.gov/sra/?term=PRJNA305211], accessions SRR2976561 to SRR2976566). A single starting cell was grown in two steps (Fig. 1 of the original paper[28]): After an initial mini-expansion, a single cell was selected as the founder for the secondary IL expansion into 20–30 cells. From this, two cells were extracted (IL-11 and IL-12) and sequenced following MDA. The remaining kindred cells from that clone were used as a bulk sequencing sample without amplification (IL-1C). IL-1C serves as the ground truth, as these cells are only very few cell divisions away from IL-11 and IL-12, and thus have almost no difference in the somatic mutations acquired. The ground truth genotype was generated using GATK HaplotypeCaller to call variant sites and bcftools mpileup to identify homozygous reference sites (with read coverage above 25 but no alternative allele present). IL-1C was only used as ground truth and not provided as input to any of the software compared here. Three more distant clones (C1, C2, C3), generated from cells after the first mini-expansion, were merged into a further bulk sample for SCcaller and ProSolo (see Software and Parameters below). Unlike other callers (all of which finished in less than 5 days), SCIPhI took 5 weeks to finish on this dataset in sensitive mode and 7.5 weeks in default mode.

Whole exome sequencing of five human granulocytes with a pedigree ground truth. For the second benchmarking dataset, blood was taken from a patient with a

constitutional mismatch repair-deficiency[48] after informed consent. Data acquisition protocols were approved by the Ethics Committee of the Medical Faculty of the Heinrich Heine University Düsseldorf. Granulocytes were selected via magnetic-activated cell sorting (MACS) and fluorescence-activated cell sorting (FACS, Fig. 4b, dataset available as EGAD00001005929 [https://ega-archive.org/datasets/EGAD00001005929] in EGA study EGAS00001004123). Individual cells were isolated using a microfluidics device and subjected to multiple displacement amplification (MDA). Using a panel of 16 primer pairs covering different genes across chromosomes (Supplementary Table 3) for quantitative real-time PCR, we selected granulocytes where at least 15 of these loci were properly amplified. For those cells, we performed whole-exome capture, sequencing library preparation, and paired-end Illumina sequencing. From the remaining sorted cell population, we also extracted bulk DNA and submitted it to whole-exome capture and paired-end Illumina sequencing without MDA, to generate a bulk background sample for ProSolo and SCcaller.

To generate the ground truth of this dataset, we could leverage previously sequenced bulk whole-exome data from the same person, their parents, and three siblings[48] (Fig. 4b). To create ground truth germline alternative allele calls, we ran three pedigree-aware variant callers (BEAGLE4.0[49], polymutt[50], and FamSeq[51,52]) and created a consensus by including only calls where all callers agree at a site and where a maximum of one caller not calling the site was allowed (Supplementary Fig. 5).

Single nucleus exome sequencing of 16 normal and 16 tumor cells from triple-negative breast cancer (TNBC)[21]. The third dataset is a published dataset from a TNBC patient, with both tumor and matched normal cells available (Fig. 4c, dataset available via the sequence read archive, project id PRJNA168068 [https://www.ncbi.nlm.nih.gov/sra/?term=PRJNA168068] or accession SRA053195). For each, 16 single cell nuclei and a bulk sample were the whole-exome sequenced. Further, a number of candidate sites for clonal and subclonal tumor mutations was confirmed by targeted duplex deep sequencing in a separate experiment[21]. We analyzed the tumor and normal cells separately, ensuring that normal cells were called with the normal bulk background sample and tumor cells with the tumor bulk background sample. As the ground truth for the tumor cells, we then used the normal bulk sample augmented with the clonal tumor mutations confirmed by targeted duplex sequencing. As the ground truth for the normal cells, we used the tumor bulk sample, removing the confirmed clonal and subclonal tumor mutations. This experimental setup aims for fairness across all competitors. Variant calling for the ground truths was performed as for cell line data above.

*Estimates of allele dropout rate validate ProSolo's model.* For the allele dropout rate, we will focus on the set of sites where the respective ground truth call is heterozygous, as these are the sites where the dropout of one of the alleles can be identified in a nonambiguous manner. More details for the three ways in which we calculate allele dropout rates are given in Supplement (Supplementary Section 2.7). Here, we give a short explanation:

1. At each ground truth heterozygous site, we sum the posterior probabilities of the two allele dropout events defined in ProSolo ("ADO to alt" and "ADO to ref" in Fig. 3d) to obtain a total allele dropout probability (Supplementary Equation S 31) and use these to compute the expected value of allele dropout across all sites. This gives us an expected allele dropout rate ("expected value" in Fig. 2, Supplementary Equation S 33).
2. We genotype all the ground truth heterozygous sites with ProSolo, take the most likely genotype, and then compare against the ground truth: heterozygous sites that ProSolo calls homozygous are counted as dropout sites. The number of such sites is divided by the total number of ground truth heterozygous sites where the respective single cell had coverage ("hom at ground truth het" in Fig. 2 and Supplementary Equation S 34).
3. We identify all heterozygous ground truth sites with a coverage of at least 7, where without amplification bias we could be reasonably sure to sample both alleles (for the reasoning see Supplement, Supplementary Section 2.7). We then count a site as an allele dropout if there is one allele (reference or alternative) with no read coverage at all, and again divide by the total number of ground truth heterozygous sites where the respective single cell had coverage ("no alt/ref read at ground truth het" in Fig. 2 and Supplementary Equation S 35).

**Reporting Summary**. Further information on research design is available in the Nature Research Reporting Summary linked to this article.

## Data availability

Lists of accessions for all samples used are given in the accompanying GitHub repository, namely at: https://github.com/ProSolo/benchmarking_prosolo/tree/v5.0/analysis_pipelines#dataset. This repository is also archived via Zenodo: https://doi.org/10.5281/zenodo.3769115. Dong et al. (2017)[28] provide the first dataset (Fig. 4a) via the Sequence Read Archive as SRA project PRJNA305211 (accessions SRR2976561 to SRR2976566): https://www.ncbi.nlm.nih.gov/sra/?term=PRJNA305211. We provide the second dataset alongside this publication (Fig. 4b) via the European Genome-Phenome Archive as EGA dataset EGAD00001005929: https://ega-archive.org/datasets/EGAD00001005929. As

this is patient data, interested researchers need to contact the data access committee of this dataset via the EGA platform and sign a license agreement to gain data access: https://ega-archive.org/dacs/EGAC00001001468. Wang et al. (2014)[21] provide the third dataset (Fig. 4c) via the Sequence Read Archive as SRA project PRJNA168068 (accession SRA053195): https://www.ncbi.nlm.nih.gov/sra/?term=PRJNA168068.

## Code availability

All code used in data analysis and preparation of the manuscript, alongside a description of necessary steps for reproducing results, can be found in a GitHub repository accompanying this manuscript: https://github.com/prosolo/benchmarking_prosolo. To ensure its long-term availability and make it citable, we have created a release of this repository and archived it on Zenodo: https://doi.org/10.5281/zenodo.3769115. Further, the code of ProSolo itself is in the following GitHub repository: https://github.com/prosolo/prosolo. The version of the code used in this study has been archived under: https://doi.org/10.5281/zenodo.5366555.

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

## Acknowledgements

This work has been supported by the Helmholtz Association, in particular through a Helmholtz Incubator grant (Sparse2Big ZT-I-0007), by the compute cluster at the Helmholtz Institute for Infection Research, and by the Katharina Hardt Stiftung. Alexander Schönhuth was supported by the Netherlands Organisation for Scientific Research (NWO: Vidi grant 639.072.309). Arndt Borkhardt and Ute Fischer were further supported by the German Federal Office for Radiation Protection (BfS) grant nos. 3618S32274 and 3618S32275.

## Author contributions

A.C.M., A.B., and U.F. conceived the original project to study single immune cells. A.C.M., A.B., U.F., and D.L. created the experimental design. U.F. obtained and processed samples. A.S. and D.L. formulated the statistical model and wrote the manuscript, with comments from A.C.M. The model was implemented by J.K. and D.L. in Rust. D.L. conducted all computational analyses, under the supervision of A.C.M. and A.S. Results were interpreted by D.L., A.C.M., and A.S. All authors edited the manuscript.

## Funding

## Competing interests

The authors declare no competing interests.
