## [Peer Review File · Nature Communications]

Title: ProSolo: Accurate and Scalable Variant Calling from Single Cell DNA Sequencing DataREVIEWER COMMENTS

Reviewer #1 (Remarks to the Author):

The manuscript by Lahnemann et al describes ProSolo, a new statistical method for detecting SNVs using single-cell deep DNA sequencing data. ProSolo achieves improved performance through modeling both MDA amplification biases and amplification errors. It further improves performance by incorporating bulking sequencing data of the same sample. The method is well presented in the manuscript and benchmarked against several existing similar algorithms. However, this reviewer has the following concerns:

1. The model developed by the authors assumes that each genomic locus has 4 alleles. This assumption may not be true for a lot of genomic regions of a tumor cell. Tumors are known to have frequent CNVs. And these CNVs can range from total deletions to very high levels of amplification. These CNVs make the allele frequencies in these loci very different from 50%. But this is not the result of MDA amplification bias.
2. The performance improvement of ProSolo comes from two sources: careful modeling and additional information from the bulking sequencing data. How much of the performance improvement comes from the mathematical model, how much from the additional information provided by the matched bulk sequencing data?
3. The manuscript states that “large enough sampling of the bulk cell population” is the source of improvement for ProSolo. How large is large enough? What is the minimum coverage depth of the bulk sample so that we can observe significant performance improvement from ProSolo? How does the change of coverage affect the performance of ProSolo? This reviewer wants a thorough evaluation of ProSolo’s performance when the coverage depth of the bulk sample changes from 20X to 400X.
4. The authors benchmarked ProSolo with bulk sequencing data against other methods which cannot take advantage of the bulk data. This may not be a fair comparison. Additional bulk sequencing requires additional cost and this additional cost brings better performance. A fair comparison will be the following example. The bulk sample is sequenced at 200X. Then additionally, 10 single cells will be sequenced at coverage depth of 20X each. ProSolo incorporates the bulk data and other methods will use the 10 additional single-cell datasets. In this example, will ProSolo still be able to outperform other methods?

Reviewer #2 (Remarks to the Author):

Variant calling from single-cell genome sequencing (SCS) data is an important problem given the possible range of application of SCS in different sub-fields of biology. To date, multiple algorithms have been developed to tackle this problem either using population-based approach (e.g., Monovar), modelling of allelic bias (e.g., SCcaller), underlying cellular lineage (e.g., SCIPhi) or read-based phasing (e.g., LiRA). This paper introduces a variant calling algorithm called ProSolo that jointly models the single

cell sample with a bulk sequencing sample from the same cell population for improving variant calling and genotyping from multiple displacement amplification (MDA)-based SCS data. At the same time, ProSolo employs an empirical model of amplification bias that allows for local variation of these artefacts.

The novelty of the paper is three-fold. First of all, it introduces a novel probabilistic model for combining bulk and SCS data for variant calling. Secondly, the variant calling method allows for a control of the false discovery rate in determining alternate allele or other events. Finally, to validate the method, the authors introduce a new whole-exome SCS dataset from human granulocytes with a pedigree ground truth. Further, the authors claim that ProSolo is the first variant caller to allow for site-specific variation of amplification error. However, this is not the case as the authors borrow the empirical model of amplification bias from a previous study (Lodato et al. 2015) and other models that infer site-specific allelic dropout have been proposed before (Hard et al. 2019).

The major strength of the method seems to come from the joint modelling of SCS and bulk data. It is important to note that this is not the first approach where bulk sequencing data is used for improving variant calling from SCS data. However, the authors present a novel comprehensive probabilistic model for joint analysis of bulk and SCS data. The model is described well and the supplementary section comprehensively describe the computation of the posterior probability of the joint events. However, the use of bulk sequencing data also raises some fundamental concerns. At the same time, I do have some practical concerns that when addressed will improve the manuscript.

Major Comments

1. The authors denote $\theta_s \in \{0, 0.5, 1\}$ as the true frequency of alternate nucleotide in the single-cell sample. When describing the single-cell events in the Table S2, there seem to be some discrepancies in the θ_s values. For example, for the event $E_{(ADO \text{ to ref})}$, since the implicit single-cell genotype is heterozygous, the value of θ_s should be 0.5, but it is written as 0. Similar discrepancy exists for other entries. More importantly, the range of values used for θ_b , the true frequency for the alternate allele in the bulk sample are not very clear for the different events. The different values of θ_b used for the different events must be clearly explained. For example, for the event $E_{(ADO \text{ to ref})}$, the authors assume the range of $\theta_b \in [0.5, 1)$, however, θ_b will actually depend on the percentage of cells carrying the alternate nucleotide. If for a somatic mutation, only 30% cells carry the heterozygous genotype, $\theta_b = 0.15$, but the event $E_{(ADO \text{ to ref})}$ is still possible in a single cell drawn from the population carrying the heterozygous genotype. Since these θ_b values are further used in calculating the posterior probabilities of single cell events, it is important to define them clearly and state the inherent assumptions if any.

2. The authors claim that a large enough sampling of the bulk cell population that the single cell comes from should contain the single cell's genotype at a particular site, unless this cell is genuinely the first cell to harbour a mutation at that site. The authors further use this for imputing the sites with no read coverage. However, imputation using this will be problematic for somatic mutation sites that are only present in a fraction of cells. In cancer, subclonal mutations can be observed in only a subset of cells and imputing such a site using the read information from the bulk sequencing data would be problematic as

an average information would be obtained. Currently, the method solely relies on bulk data for imputation and assumes all single cell genotypes to be equally likely. This can be improved by assuming a prior distribution on single-cell genotypes that can vary from one site to another. The prior can also be designed by considering the data of the other single cells at that site and the variant allele frequency from the bulk data. The imputation accuracy of ProSolo can be compared against that of SCIPhi by considering simulated datasets containing varying levels of missing reads. Third-party unbiased single-cell sequencing simulators (e.g., CellCoal by David Posada 2020) are available now.

3. While ProSolo has been validated on whole-genome and whole-exome sequencing datasets, the used datasets contain very small number of cells that are drawn from a likely homogeneous cell population. A major application of SCS data is in elucidating heterogeneous cell population such as in cancer. The use of bulk dataset for a heterogeneous cell population can have some confounding effect in variant calling. ProSolo should be applied on a cancer SCS dataset (e.g., Wang et al 2014 Nature, Leung et al 2017 Genome Research) to analyse its ability in variant calling from a heterogeneous cell population. This experiment will also be helpful in identifying the situations where using bulk data might have advantages or disadvantages. While direct validation may be difficult for such a dataset, ProSolo's results can be compared against SCIPhi or SCAN-SNV, methods that are optimized for detection of somatic SNVs from heterogeneous cell populations.

4. The absolute recall of ProSolo and other algorithms is low for the granulocyte dataset and the authors argue that lower coverage of the background bulk sample might be one reason. This also leads to an important question regarding how results of ProSolo can vary based on the coverage of the bulk sample. A downsampling experiment can be performed to assess this scenario. Alternatively, simulation experiments can be designed based on existing simulators.

5. The authors found that SCIPhi was the closest competitor in terms of precision and recall for the granulocyte dataset but took very long time (3 weeks) to run. Another method scVILP (Edrisi et al. WABI 2019) is already available that performs variant calling by considering the underlying phylogeny of the cells (just like SCIPhi) and achieves F1 score similar or better than SCIPhi but significantly reduces runtime by employing a combinatorial optimization approach. The authors should compare ProSolo's results against that of SCIPhi for the granulocyte dataset.

Minor Comments

1. The manuscript contains many statements/claims that need to be qualified. For example, the authors claim that "ProSolo is the first method for SNV calling from MDA single cell sequencing data to comprehensively model both amplification bias and amplification errors in a way that allows for site-specific variation". This statement needs to be modified as stated above. Further the authors mention that "The most precise single cell variant callers to date, SCcaller and SCIPhi". This is not universally true as has been observed in previous studies. There are more such statements that need to be rewritten.

2. In estimating the allelic dropout rate using ProSolo, ProSolo's estimate by computing the fraction of

heterozygous sites that were called homozygous by ProSolo was slightly higher (for granulocyte dataset) than the expected allele dropout rate calculated based on posterior probability. Does it actually mean that ProSolo was unable to correct for the sites that were affected by allelic dropout? The authors can discuss on this more to clarify the results.

Reviewer #3 (Remarks to the Author):

The manuscript by Laehnemann and coauthors describes a variant caller for single cells, informed by bulk samples. While clearly an improvement on current methods in the real data evaluations, the setting of calling mutations in a handful of single cells (informed by the bulk) is rather niche, whereas practical interest in single-cell sequencing is in lineage tracing, as also highlighted in the motivation.

Calling germline mutations in single-cells therefore does not seem very useful, it is precisely the somatic mutations which inform the cell lineage trees and are important for understanding the evolution of tumors for example. However, this is the setting which the different tools are benchmarked.

Here there are several issues with the use of SCIPhi. First this tool returns the posterior probability of mutational presence, which could be used to generate a full ROC curve. Second the runtimes seem excessive. For the case where there are two cells, there is only one possible tree, so the inference only needs to learn the parameters of the beta-binomial models (this could actually be done numerically very quickly), so the number of iterations would not need to be high (and not need to include any tree moves). The runtime grows with the number of mutations included, which is why germline mutations (from the bulk) are normally filtered out to focus on the informative data for tree lineage building.

In terms of methodological advance, the beta-binomial model is (naturally) similar to other approaches from Lodato et al onwards, with novelty here a coverage-dependent drop-out term, and bulk contribution to the likelihood. The claim of being the first tool to offer FDR control is based on ProSolo returning posterior probabilities, but these are generated by others methods (SCIPhi for example).

In the methods, the parameters defining the distributions should be described (and not relegated to the supplement). Here the authors seem to use fixed values, but they should be learned from the data as part of the mutation calling (and with a prior to obtain posterior mutation calls).

To summarize, while the advances described by Laehnemann and colleagues for improved read count models for individual cells are welcome, to have a wider impact the method should include parameter estimation as well as sharing information across cells according to their phylogenetic relationships.

Response to peer reviews of “ProSolo: Accurate and Scalable Variant Calling from Single Cell DNA Sequencing Data”

General Remarks

We would like to thank all reviewers for their thorough and insightful reviews. In the following, we respond to all questions and comments in detail. Attached to these responses are the main manuscript and the supplement with changes highlighted, to allow for quick reference to changes by page and line number (numbering restarts for both the main manuscript and the supplement).

Beforehand, we would like to point out that we realized that bcftools version 1.6 included a bug that parsed the ProSolo output incorrectly. Updating bcftools to the more recent, bug-free version 1.10, and re-generating all affected results, ProSolo’s recall improved for false discovery rates of 0.05 and greater. We have adapted the manuscript accordingly.

Point-by-point response to reviewer comments

Reviewer #1 (Remarks to the Author):

> The manuscript by Lahnemann et al describes ProSolo, a new statistical method for detecting SNVs using single-cell deep DNA sequencing data. ProSolo achieves improved performance through modeling both MDA amplification biases and amplification errors. It further improves performance by incorporating bulking sequencing data of the same sample. The method is well presented in the manuscript and benchmarked against several existing similar algorithms.

We thank the reviewer for acknowledging key points and achievements of our work.

> However, this reviewer has the following concerns:

> 1. The model developed by the authors assumes that each genomic locus has 4 alleles. This assumption may not be true for a lot of genomic regions of a tumor cell. Tumors are known to have frequent CNVs. And these CNVs can range from total deletions to very high levels of amplification. These CNVs make the allele frequencies in these loci very different from 50%. But this is not the result of MDA amplification bias.

We fully agree with the reviewer that copy number variation is a very important issue in the analysis of single tumor cells, and that integrating SNV and CNV analysis will surely improve variation calling results.

At this point in time in research, however, combining the analysis of copy number variation with the analysis of single nucleotide variants in single cells is a methodical challenge that seems to exceed the range of what is currently possible (see the 4 references following this response for an overview of the state of the art). Currently, basic calling and genotyping of SNV's for single cells still poses major issues, due to the statistical uncertainties associated with this. Introducing CNV's adds another level of statistical uncertainty, because, as the reviewer points out, copy numbers can span a relatively large range.

We argue that the uncertainties introduced by CNV's can only be dealt with if the basic issues have been resolved. In that sense our work can be considered as a step in that direction -- resolving statistical uncertainties, step by step, where each step requires substantial statistical considerations.

Note as well that benchmarking such an integrated tool would require a whole genome single cell sequencing dataset from a cancer sample with a well-characterized ground truth of both CNV's and SNV's, created from data separate from the one used for variation calling. We are not aware that such datasets exist.

References:

- Bakker B, Taudt A, Belderbos ME, Porubsky D, Spierings DCJ, de Jong TV, Halsema N, Kazemier HG, Hoekstra-Wakker K, Bradley A, de Bont ESJM, van den Berg A, Guryev V, Lansdorp PM, Colomé-Tatché M, Foijer F. Single-cell sequencing reveals karyotype heterogeneity in murine and human malignancies. *Genome Biol.* 2016;17:115. <https://doi.org/10.1186/s13059-016-0971-7>.
- Fan J, Lee H-O, Lee S, Ryu D-E, Lee S, Xue C, Kim SJ, Kim K, Barkas N, Park PJ, Park W-Y, Kharchenko PV. Linking transcriptional and genetic tumor heterogeneity through allele analysis of single-cell RNA-seq data. *Genome Res.* 2018;28(8):1217–27
- Garvin T, Aboukhalil R, Kendall J, Baslan T, Atwal GS, Hicks J, Wigler M, Schatz MC. Interactive analysis and assessment of single-cell copy-number variations. *Nat Methods.* 2015;12(11):1058–60.
- Patel AP, Tirosh I, Trombetta JJ, Shalek AK, Gillespie SM, Wakimoto H, Cahill DP, Nahed BV, Curry WT, Martuza RL, Louis DN, Rozenblatt-Rosen O, Suvà ML, Regev A, Bernstein BE. Single-cell RNA-seq highlights intratumoral heterogeneity in primary glioblastoma. *Science.* 2014;344(6190):1396–401.

> 2. The performance improvement of ProSolo comes from two sources: careful modeling and additional information from the bulking sequencing data. How much of the performance

improvement comes from the mathematical model, how much from the additional information provided by the matched bulk sequencing data?

We agree that this is an interesting question. An entirely clear answer is difficult to provide, as the integration of the bulk is part of the mathematical model.

To provide as much insight as possible, we performed the following downsampling experiment (see also point 3 below): we gradually decreased the coverage of the bulk to zero (in the granulocyte whole exome dataset, more details below in the answer to question 3), and tracked the resulting performance rates.

This revealed that the probabilities that ProSolo works with for controlling the FDR became gradually less informative, reflecting an effect that was to be expected. Eventually (at coverage zero) controlling false discoveries at a rate of less than 0.2 will lead to zero recall (in the granulocyte whole exome dataset), which means that the certainty of all calls dropped below a probability of 0.8 (new Supplementary Figure S11 and accompanying text in Supplementary Section S2.4). Still, for false discovery rates greater than or equal 0.2, operable recall can be achieved, while precision reflects the decreased FDR one wishes to control (hence drops below 0.8).

In summary, ProSolo is fully operable for bulk coverage of as low as 4X (or greater, of course). At bulk coverage of 0X (no bulk), ProSolo preserves recall, but is no longer able to enable sufficiently fine grained FDR control.

> 3. The manuscript states that "large enough sampling of the bulk cell population" is the source of improvement for ProSolo. How large is large enough? What is the minimum coverage depth of the bulk sample so that we can observe significant performance improvement from ProSolo? How does the change of coverage affect the performance of ProSolo? This reviewer wants a thorough evaluation of ProSolo's performance when the coverage depth of the bulk sample changes from 20X to 400X.

We agree with the reviewer that guidance regarding the necessary coverage in the bulk sample for a good performance of ProSolo is imperative and thank for this comment.

To provide this information, we performed the downsampling experiment on the whole exome sequencing dataset consisting of five granulocytes. See also point 2 above, where we discussed these experiments, and stressed what happens if bulk coverage drops to 0X.

We reduced the average coverage in the bulk sample from the original 40X down to 28X, 16X, 4X and 0X, respectively, and then called variants with ProSolo. The analysis shows that ProSolo's model requires a bulk sample to provide a sufficiently fine grained control of FDR (see also response to point 2 above). But already a bulk sample coverage of 4X re-establishes the full operability of ProSolo's FDR control, because ProSolo's probabilities have returned to being

fully informative. Further increasing coverage to 16X, 28X or 40X does not lead to improvements that are as substantial as the improvement seen when going from 0X to 4X.

As a consequence of fully operable FDR control already at 4X, one also observes that fully operable precision is achieved as well. See new Supplementary Figure S11 and accompanying text in Supplementary Section S2.4, as well as the integration of these findings in the Discussion (changes highlighted main manuscript, page 24, lines 466-469).

The analysis of another dataset suggested by Reviewer #2 (Wang et al. 2014) corroborates these findings: uncertainties remain too large when calling somatic variants that correspond to subclonal levels that the bulk can no longer capture. As pointed out by the experiments from above, when aiming at a particular subclonal variant, the obvious remedy is to sequence the bulk at a coverage that samples at least one good sequencing read from the alternative allele of the corresponding subclone---for a heterozygous alternative subclone at a frequency of 0.01, the alternative allele should occur at frequency 0.005 and the bulk should accordingly be sequenced at a coverage above 200X ($= 1 / 0.005$). For some additional details please also see the answer to point 3 of Reviewer #2 and the newly added section S 2.5 in the Supplement.

> 4. The authors benchmarked ProSolo with bulk sequencing data against other methods which cannot take advantage of the bulk data. This may not be a fair comparison. Additional bulk sequencing requires additional cost and this additional cost brings better performance. A fair comparison will be the following example. The bulk sample is sequenced at 200X. Then additionally, 10 single cells will be sequenced at coverage depth of 20X each. ProSolo incorporates the bulk data and other methods will use the 10 additional single-cell datasets. In this example, will ProSolo still be able to outperform other methods?

We agree with the reviewer that a fair comparison between tools with different approaches is challenging. We do believe, however, that our analyses are already fair in the sense that the reviewer points to here, because it is debatable whether ProSolo requires more data than the other approaches. Take our whole exome dataset with five granulocytes as an example. The different samples (PAGX single cells; PNG is bulk) have the following file sizes after mapping of reads:

- PAG10: 6.8G
- PAG1: 7.6G
- PAG2: 5.7G
- PAG5: 6.6G
- PAG9: 6.7G
- PNG: 9.1G

Thus, the joint data of the single cell samples (PAG) is 33.4G, while the coverage of the bulk data (PNG) corresponds to 9.1G. Thus, every calling of ProSolo on a single cell with the bulk background uses at most 16.7G of read data (PAG1 + PNG), while any tool that uses all single

cells in combination (e.g. MonoVar, SCIPhi) is provided with 33.4G of read data. This means that ProSolo outperforms its toughest competitors at at most 16.7G of read data in comparison to 33.4G (by coincidence = 2 x 16.7G) required by the competitors.

From that perspective, ProSolo outperforms other tools at lower sequencing costs for analysing few cells and for larger numbers of cells, the additional cost due to sequencing a bulk will become less pronounced.

We therefore argue that it is perfectly justified to say that no tool has considerably greater sequencing costs in comparison with others, to achieve competitive performance. Furthermore, while more coverage will usually mean higher sequencing costs, a single cell sequencing sample comes with additional costs for sorting, lysing and whole genome amplifying the single cell DNA, rendering additional single cell samples more expensive than a single added bulk sequencing sample.

Reviewer #2 (Remarks to the Author):

> Variant calling from single-cell genome sequencing (SCS) data is an important problem given the possible range of application of SCS in different sub-fields of biology. To date, multiple algorithms have been developed to tackle this problem either using population-based approach (e.g., Monovar), modelling of allelic bias (e.g., SCcaller), underlying cellular lineage (e.g., SCIPhi) or read-based phasing (e.g., LiRA). This paper introduces a variant calling algorithm called ProSolo that jointly models the single cell sample with a bulk sequencing sample from the same cell population for improving variant calling and genotyping from multiple displacement amplification (MDA)-based SCS data. At the same time, ProSolo employs an empirical model of amplification bias that allows for local variation of these artefacts.

> The novelty of the paper is three-fold. First of all, it introduces a novel probabilistic model for combining bulk and SCS data for variant calling. Secondly, the variant calling method allows for a control of the false discovery rate in determining alternate allele or other events. Finally, to validate the method, the authors introduce a new whole-exome SCS dataset from human granulocytes with a pedigree ground truth. Further, the authors claim that ProSolo is the first variant caller to allow for site-specific variation of amplification error. However, this is not the case as the authors borrow the empirical model of amplification bias from a previous study (Lodato et al. 2015) and other models that infer site-specific allelic dropout have been proposed before (Hard et al. 2019).

We thank the reviewer for this knowledgeable assessment and pointer. As for the last point, we have missed to point out our achievements sufficiently clearly. We thank the reviewer and have adapted the manuscript in an attempt to eliminate this misunderstanding: while we found the last paragraph of the Introduction to provide the correct statements, the Abstract had to be

mended (changes highlighted main manuscript, page 2, lines 29-31), such that both passages make the appropriate statements.

Specifically, we fully acknowledge Lodato et al. (2015) and Hard et al. (2019) as studies that precede ours and that address amplification-related, site-specific biases. As noted by the reviewer, we even make use of the coverage-specific probability distributions fitted and suggested in Lodato et al. (2015) ourselves.

However, *Lodato et al. (2015) do not describe a variant caller that is based on the basic site-specific statistics they introduced. **Our achievement is to do just this, and integrate this further into a comprehensive statistical model of all biases and errors affecting the calling and genotyping of variants in single cells.*** This also includes the integration of a bulk sequencing sample.

Within this context, we further realized that some other key strengths of our model were not stated sufficiently clearly, and have amended this throughout the manuscript. One of these particular strengths of the model is to enable *efficient computations of accurate probabilities* of variants to be present or not--while other approaches (e.g. SCIPhI) make use of runtime-intense approximations of these probabilities.

A further key achievement is to enable operable false discovery control across the whole range of relevant thresholds of control, and as a consequence, provide a very high recall at utmost precision. No other tool is able to provide truly functional FDR control. In addition, the required wall clock runtime of the only competitive tool, SCIPhI, will be substantially higher (requiring weeks, while we need only hours or days), if similarly good recall and precision need to be provided.

In particular the last point illustrates why computational efficiency is a key issue in single cell variant calling. Achieving computational efficiency required substantial additional statistical modeling. Compared with earlier work, *we are the first to describe a sufficiently rigorous and encompassing statistical model, which furthermore allows for a computationally efficient evaluation that will scale with the rapidly increasing throughput of droplet-based single cell whole genome amplification techniques and the datasets growing accordingly.* To especially highlight this aspect of scalability, we also included it in the title of the manuscript.

Hard et al. (2019), while integrating amplification related biases in a site-specific way, do not describe or make use of a rigorous and encompassing statistical model as ours to take care of these biases. In Hard et al. (2019), please see equations (1) and (2) in the section 'Stats', where Hard et al. suggest ad-hoc (heuristic) procedures for dealing with individual polymorphic sites.

In summary, the statistical model underlying ProSolo is the first to

- be encompassing, in the sense that it addresses all (both nonspecific and single cell sequencing specific) errors and biases that affect the calling and genotyping of variants in single cells in a site-specific manner.
- deals with all these errors in a scalable, because computationally efficient, manner.
- allow for a meaningful control of the false discovery rate.

The combination of these 3 points, and especially the scalability, has been considered a notorious statistical and computational challenge already in bulk settings (see e.g. <https://doi.org/10.1186/s13059-020-01993-6>). The reason is that straightforward approaches to dealing with site-specific biases in terms of fully statistical (that is non ad-hoc) approaches require runtimes exponential in the coverage. Here, we present an algorithm that requires runtime linear in the number of reads that cover a site, which is the fastest way conceivable.

> The major strength of the method seems to come from the joint modelling of SCS and bulk data. It is important to note that this is not the first approach where bulk sequencing data is used for improving variant calling from SCS data. However, the authors present a novel comprehensive probabilistic model for joint analysis of bulk and SCS data. The model is described well and the supplementary section comprehensively describe the computation of the posterior probability of the joint events. However, the use of bulk sequencing data also raises some fundamental concerns.

Again, we would like to thank the reviewer for acknowledging potential strengths of ProSolo and the detailed consideration of the technical parts. We further analyse and discuss different aspects of using a bulk sample in the answers to major comment 3 of this reviewer and major comments 2 and 3 of Reviewer #1, and would like to point to further strengths of ProSolo's statistical method as mentioned in the answer right above.

At the same time, I do have some practical concerns that when addressed will improve the manuscript.

Major Comments

> 1. The authors denote θ_s in $\{0, 0.5, 1\}$ as the true frequency of alternate nucleotide in the single-cell sample. When describing the single-cell events in the Table S2, there seem to be some discrepancies in the θ_s values. For example, for the event $E_-(ADO \text{ to } ref)$, since the implicit single-cell genotype is heterozygous, the value of θ_s should be 0.5, but it is written as 0. Similar discrepancy exists for other entries. More importantly, the range of values used for θ_b , the true frequency for the alternate allele in the bulk sample are not very clear for the different events. The different values of θ_b used for the different events must be clearly explained. For example, for the event $E_-(ADO \text{ to } ref)$, the authors assume the range of θ_b in $[0.5, 1)$, however, θ_b will actually depend on the percentage of cells carrying the alternate nucleotide. If for a somatic mutation, only 30% cells carry the heterozygous genotype,

theta_b=0.15, but the event E_(ADO to ref) is still possible in a single cell drawn from the population carrying the heterozygous genotype. Since these theta_b values are further used in calculating the posterior probabilities of single cell events, it is important to define them clearly and state the inherent assumptions if any.

We thank the reviewer for pointing out an issue where our writing was not clear enough, and for spotting these ambiguities at this level of detail. The issue was that the legend of Figure 1D was flawed and the text describing it imprecise: the y-axis and the x-axis were meant to refer to likelihood density estimates raised by ProSolo, and not the hidden variables that capture the truth. While the truth does not know conflicts, ProSolo's likelihood density estimates can be conflicting and assumptions in resolving them need to be explained clearly. We mended the issue by exchanging the Figure 1D legend θ_s and θ_b , which refer to the true allele frequencies, by $\tilde{\theta}_s$ and $\tilde{\theta}_b$, which are meant to refer to the likelihood density estimates across the respective allelic frequencies as raised by ProSolo (changes highlighted main manuscript, page 6). We also adjusted the labels of Figure S 2 and Table S 1 in the supplement accordingly (changes highlighted supplement, pages 16 and 17). Finally, we clearly explain the assumptions about bulk cell population mixtures in the accompanying text in both the main manuscript and the supplement (changes highlighted main manuscript, page 10, lines 200-216; changes highlighted supplement, page 17, lines 257-271). In more detail:

Where ProSolo's estimates for bulk and single cell samples are in conflict (which may happen in rare cases), we need to resolve the contradictory scenario. In that case, we need to make choices, because we lack statistical handles to resolve the contradictions. Both Figure 1D and Table S1 (in the supplement) illustrate our choices.

As for the choices themselves, we make two important assumptions that we now further highlight in the Methods section of the main manuscript:

- (i) We assume that the bulk reflects a mixture of at most 2 cell subpopulations with regard to any genomic site (illustrated in the lower panel of Figure 1 D). Either cells that are homozygous for reference are mixed with heterozygous cells, or cells that are homozygous for the alternative allele are mixed with heterozygous cells, which is a reasonable, because most parsimonious assumption in practice.
- (ii) For the likelihood density estimate of a given combination of single cell and bulk allele frequencies, we trust the single cell allele frequency (and respective genotype classification) only if it is deemed to be present in the bulk at the given bulk frequency (according to the above assumption about subpopulations).

By these assumptions, the interpretation of the likelihood density estimate for $\tilde{\theta}_b < 0.5$ is that the bulk consists of only one subpopulation, namely all cells are heterozygous. We also reason that $\tilde{\theta}_s = 0$ is more likely due to a cell that is homozygous for the reference than allele dropout (both cases cannot be distinguished, when considering the single cell alone). This reasoning explains our choices:

Whenever there is reason to believe that the bulk harbors cells that are homozygous for reference, which is reflected by $\tilde{\theta}_b < 0.5$, we interpret $\tilde{\theta}_s = 0$ as homozygous for reference ('hom ref'). On the other hand, $\tilde{\theta}_b \geq 0.5$ implies that the bulk does not contain cells that are homozygous for the reference. In that case, we interpret the case $\tilde{\theta}_s = 0$ as an allele dropout ('ADO to ref'). $\tilde{\theta}_b \geq 1$ finally means that all cells in the bulk are homozygous for the alternative allele. The most likely interpretation for $\tilde{\theta}_s = 0$ or 0.5 then are amplification errors, which we call accordingly ('err ref').

Note that analogous reasoning applies for the events where 'ref' is replaced with 'alt'.

As mentioned throughout, notation in the manuscript has now been adapted to the tilde-notation and the respective text introduces this as notation for ProSolo's likelihood density estimates over allele frequencies. In addition, the assumptions about the bulk cell population and that the bulk needs to contain a subpopulation with the single cell genotype for it to be trusted, are now stated in the main manuscript's text, while this was only mentioned in a figure caption, previously.

> 2. The authors claim that a large enough sampling of the bulk cell population that the single cell comes from should contain the single cell's genotype at a particular site, unless this cell is genuinely the first cell to harbour a mutation at that site. The authors further use this for imputing the sites with no read coverage. However, imputation using this will be problematic for somatic mutation sites that are only present in a fraction of cells. In cancer, subclonal mutations can be observed in only a subset of cells and imputing such a site using the read information from the bulk sequencing data would be problematic as an average information would be obtained. Currently, the method solely relies on bulk data for imputation and assumes all single cell genotypes to be equally likely. This can be improved by assuming a prior distribution on single-cell genotypes that can vary from one site to another. The prior can also be designed by considering the data of the other single cells at that site and the variant allele frequency from the bulk data. The imputation accuracy of ProSolo can be compared against that of SCIPhi by considering simulated datasets containing varying levels of missing reads. Third-party unbiased single-cell sequencing simulators (e.g., CellCoal by David Posada 2020) are available now.

We agree with the reviewer and thank him for this interesting suggestion. Integrating further reasonable assumptions on how to impute missing allele frequencies at insufficiently covered sites may be a means to further boost performance for the method.

As it stands, we see the following benefits of the current model:

(i) Without making use of imputation beyond the basic bulk assisted corroboration of calls, our method outperforms other methods that make more sophisticated assumptions on imputation of

missing alleles. We therefore argue that our core model has substantial advantages over the core models that other methods work with.

While the integration of imputation across cells in our model would be possible, this would limit the scalability of the method. Namely, when considering the alignment files of all single cells at once, this will hit limits as to how many single cells you can process in a dataset. Note that *it is a very traditional issue in bioinformatics that the exploration of phylogenetic trees alongside with other tasks is prone to lead to exploding runtimes because of the super-exponential growth of the number of possible tree topologies, which severely restricts the computational efficiency when using such approaches in probabilistic modelling*, and puts the immediate integration of tree based imputation into single cell variant calling into a larger context.

Instead, we would like to take a decidedly different approach. In particular:

(ii) Our model does not require any additional knowledge-driven assumptions across cells to work on a single cell, but focuses on exploiting the data to a maximal degree. In doing so, it captures the available data and its uncertainties per cell---and allows integrating across different cells downstream, for example via phylogenetic reconstruction that can take into account genotype probabilities and the uncertainties that they capture. This will yield an estimation of a tree topology, that could in turn be used to improve the genotype probabilities based on the phylogenetic constraints.

This is similar to what SCIPHI does, but SCIPHI's integration across single cells directly from aligned sequencing reads generates intermediate (mpileup) files in the size range of hundreds of Gigabytes for the datasets in this manuscript, reflecting the theoretical issues pointed out above. For larger cell numbers, this can be prohibitive. Instead, we suggest to first capture the data and its uncertainties as well as possible, *before* introducing additional assumptions across cells, and thus reducing the sequencing data to genotype likelihoods per site, first. The integration across cells should then be left to methods that specialize on reconstructing phylogenies in an uncertainty-aware manner.

With this considered, we would recommend:

(iii) To capture the data and its uncertainties in the best way possible, users should not impute sites in a single cell that have no read coverage at all, unless this would severely limit downstream analyses (e.g. because a downstream tool cannot handle missing data). Instead, they should pass down the information about missing data in the single cells to downstream analyses that can then integrate across cells. Wherever a single cell has coverage, even if it is only a single read, that information (with all the uncertainties from modelling amplification bias) will be useful for downstream integration.

We have accordingly amended the Discussion with a detailed paragraph about imputation and hope that this addresses the reviewer's concerns with the points made here (changes highlighted main manuscript, page 25, lines 496-501).

Finally, we also looked at CellCoal for simulating single cells. However, the focus of CellCoal is on simulating input data for phylogenetic lineage reconstruction and thus only outputs the input for such methods (which we consider downstream of ProSolo), namely variants in VCF files. On this variant data, we obviously cannot call variants but instead require aligned sequencing reads as input. To generate such data would in turn require a read simulator that both integrates variants and respects (site-specific) allelic frequencies. Especially for the latter, we did not find a read simulator that can do this.

> 3. While ProSolo has been validated on whole-genome and whole-exome sequencing datasets, the used datasets contain very small number of cells that are drawn from a likely homogeneous cell population. A major application of SCS data is in elucidating heterogeneous cell population such as in cancer. The use of bulk dataset for a heterogeneous cell population can have some confounding effect in variant calling. ProSolo should be applied on a cancer SCS dataset (e.g., Wang et al 2014 Nature, Leung et al 2017 Genome Research) to analyse its ability in variant calling from a heterogeneous cell population. This experiment will also be helpful in identifying the situations where using bulk data might have advantages or disadvantages. While direct validation may be difficult for such a dataset, ProSolo's results can be compared against SCIPhi or SCAN-SNV, methods that are optimized for detection of somatic SNVs from heterogeneous cell populations.

We clearly agree about the major application of SCS data in cancer research and the interest in a larger number of cells. We thank the reviewer for the suggestions and have now included part of the data from Wang et al. (2014) that comes with good possibilities for using and generating ground truths in a way that reflects fairness for all participating competitors, which was unclear how to do for the Leung et al. (2017) data.

For the general determination of ground truth genotypes, we could use the tumor bulk sample for validating calls from the normal single cells together with the normal bulk, and vice versa. In addition, we could augment these ground truths by respectively removing and adding the known non-synonymous somatic mutations that Wang et al. (2014) validated with a separate deep sequencing of targeted duplex bulk DNA (new panel C in Figure 2, changes highlighted main manuscript page 16, lines 307-318). In addition, these validated somatic variants could be used to determine a clonal and subclonal recall for all the software in the benchmarking.

The only issue remaining in the benchmarking setup disfavors ProSolo: we expect a bulk coverage of 35X to be too low to establish full FDR control (and hence superior recall) for somatic variants where the alternative allele is present at a frequency below about $0.029 = 1 / 35$ (thus, for heterozygous somatic variants, a subclonal frequency of $0.058 = 0.029 * 2$ is the minimum we expect to be able to call), simply because the respective allele will not be sampled

in the bulk sequencing experiment. At the same time, competitors profit from the coverage accumulated by joining 16 single cells, which exceeds 35X in total. Please note however that current tumor bulk sequencing easily exceeds 100X, especially when looking only at the exome, at reasonable cost.

The general analysis of precision and recall on this dataset confirms our conclusions for ProSolo drawn so far (new panel C in Figure 3, changes highlighted main manuscript page 19, lines 362-372). Beyond this, it now also demonstrates how SCIPhI operates on greater amounts of single cells. In this regard, SCIPhI meets the expectations: because of enhanced imputation, it achieves recall rates that exceed the ones of all competitors, including ProSolo. At the same time, because of the substantially larger space of phylogenetic trees to be explored for imputation, SCIPhI requires runtimes that exceed those of the competitors quite substantially.

In some more detail:

SCIPhI seems to benefit especially from cross-cell imputation of data across cells, when cell populations are homogeneous (reflected by greater recall on normal, not tumor, single cells). However, analyzing the recall of the somatic variants validated by Wang et al. (2014) demonstrates that SCIPhI over-imputes some somatic mutations (new Supplementary Figure S 12, new Supplementary Section S 2.5, changes highlighted supplement, page 43, lines 649-680).

At the same time, the analysis provides further guidance on the usage of ProSolo. On the heterozygous alternative genotypes in particular, the recall of ProSolo for somatic variants is lower than that for most other tools on the heterozygous alternative genotypes, especially for subclonal somatic mutations (new Supplementary Figure S 12). As already alluded to above, we think that this is attributable to the relatively low bulk coverage of around 35X, at which we would expect alleles present at a frequency above 2.9% (= 1/35) to be sampled in the bulk. Note that 52 of the 94 validated subclonal somatic mutations fall below the threshold of 2.8%. Without any allele evidence in the bulk, our model considers these alleles erroneous. The simple remedy would be to increase the bulk coverage to the desired level of sensitivity, a requirement that should already be met by current-day tumor bulk sequencing data that usually comes at > 100X; this should drastically lift ProSolo's recall of subclonal somatic variants.

[As a side remark, consider that an optimal sampling for the bulk would include cells from different parts of the tumor to cover as many subclones as possible. If this is given, ProSolo should even have advantages over approaches that rely on consensus across multiple single cells, for example when a subclone is represented by only one sampled cell, but cells from the same subclone were also sampled for the bulk sequencing.]

> 4. The absolute recall of ProSolo and other algorithms is low for the granulocyte dataset and the authors argue that lower coverage of the background bulk sample might be one reason. This also leads to an important question regarding how results of ProSolo can vary based on

the coverage of the bulk sample. A downsampling experiment can be performed to assess this scenario. Alternatively, simulation experiments can be designed based on existing simulators.

While we agree in general that lower bulk coverage does not lead to improvements, we are not quite sure whether we claimed that the low coverage of the background bulk for the granulocyte dataset is the reason for the low recall. We rather believe that the low recall is due to the limited coverage in the single cells themselves. Maybe there was a misunderstanding?

Nevertheless, and also following suggestions from Reviewer #1, we now include a downsampling experiment on the whole exome sequencing dataset. Respective results are presented in the Supplement, downsampling the bulk from 40X in several steps down to 4X and, eventually, 0X. We see that while lowering bulk coverage indeed leads to losses, reductions in terms of overall performance are rather negligible. At 0X, ProSolo still seems to be competitive in terms of recall. False discovery rate control (FDR) however, which is still fully functional at 4X, does no longer work sufficiently at 0X. The newly added somatic variant recall analysis, which shows that ProSolo's recall of subclonal variants obviously suffers from bulk coverage that is too low to cover subclonal variants seem to corroborate this idea.

But this also immediately suggests the remedy for ProSolo's performance on this type of variant of high interest to researchers: ProSolo's advantages can be optimally exploited when raising tumor bulk coverage to at least 100X, and when ensuring to sample bulk from different subclones (by spatially distributed sampling).

> 5. The authors found that SCIPhi was the closest competitor in terms of precision and recall for the granulocyte dataset but took very long time (3 weeks) to run. Another method scVILP (Edrisi et al. WABI 2019) is already available that performs variant calling by considering the underlying phylogeny of the cells (just like SCIPhi) and achieves F1 score similar or better than SCIPhi but significantly reduces runtime by employing a combinatorial optimization approach. The authors should compare ProSolo's results against that of SCIPhi for the granulocyte dataset.

We thank the reviewer for pointing to this new tool that we had not yet come across. We would have very much welcomed a quicker alternative to SCIPhi, with similar accuracy to SCIPhi, because while on the one hand, SCIPhi is the toughest competitor, it is also (by far) the slowest competitor (as the reviewer points out, as well). SCIPhi runs on the data in this manuscript require weeks, whereas ProSolo requires days.

We thus tried to run scVILP on all three datasets in our benchmarking. First, it became obvious that scVILP is not yet production-quality software. After resolving initial installation problems (see: <https://github.com/mae6/scVILP/pull/4>) and issues at runtime (see: <https://github.com/mae6/scVILP/pull/2> and <https://github.com/mae6/scVILP/pull/1>), we consistently ran into "out of memory" errors. These errors can be assigned to the integer linear

programming solver Gurobi that scVILP uses. The solver throws these errors even when providing more than a Terabyte of RAM.

None of the competitors experience such issues, so we conclude that scVILP reflects prototype style proof-of-concept software that only was able to run on some selected, exemplary datasets.

We documented this in detail in scVILP's GitHub repository, to bring this to the attention of the original developers (see: <https://github.com/mae6/scVILP/issues/5>).

In our paper, we document our attempts and the resulting error in our Supplementary Section S 2.2 Software and Parameters (changes highlighted supplement, page 34, lines 536-546).

Minor Comments

> 1. The manuscript contains many statements/claims that need to be qualified. For example, the authors claim that "ProSolo is the first method for SNV calling from MDA single cell sequencing data to comprehensively model both amplification bias and amplification errors in a way that allows for site-specific variation". This statement needs to be modified as stated above. Further the authors mention that "The most precise single cell variant callers to date, SCcaller and SCIPhI". This is not universally true as has been observed in previous studies. There are more such statements that need to be rewritten.

As pointed out above, the claim, as drawn from the Abstract, is not entirely correct indeed (while we had the unambiguous claim at the end of the Introduction). We now have corrected this. Therefore, we stand by the (now consistently cited) claim that we account for site-specific biases and errors in a statistically consistent as well as in a computationally efficient, hence scalable manner.

Regarding the second mentioned claim, we are not aware of any studies demonstrating to outperform SCcaller and SCIPhI. We would be glad for further pointers by the reviewer and would of course adjust the respective statement if necessary.

> 2. In estimating the allelic dropout rate using ProSolo, ProSolo's estimate by computing the fraction of heterozygous sites that were called homozygous by ProSolo was slightly higher (for granulocyte dataset) than the expected allele dropout rate calculated based on posterior probability. Does it actually mean that ProSolo was unable to correct for the sites that were affected by allelic dropout? The authors can discuss on this more to clarify the results.

Thanks for pointing this out. Interestingly, as we also mention in the allele dropout paragraph of the discussion section in the main manuscript, the relation is reversed in the whole genome cell line data. As we discuss there, this indicates that the amplification bias distributions suggested

by Lodato et al. (2015), which we here assume to be applicable in general, may not be optimal for all datasets. Instead, these differences in the estimations can probably only be resolved by dataset-specific distributions that reflect that library and amplification protocols vary for each dataset. We thus mention the estimation of such dataset-specific distributions as future work and expect the results of our model to improve further.

Reviewer #3 (Remarks to the Author):

> The manuscript by Laehnemann and coauthors describes a variant caller for single cells, informed by bulk samples. While clearly an improvement on current methods in the real data evaluations, the setting of calling mutations in a handful of single cells (informed by the bulk) is rather niche, whereas practical interest in single-cell sequencing is in lineage tracing, as also highlighted in the motivation.

We agree with reviewer #3 that lineage tracing is a predominant (if not *the* predominant) concern in the analysis of single cell sequencing. We equally agree that the analysis of more than just a handful of single cells is a very relevant setting. We now include an additional dataset with a greater number of single cells (Wang et al., 2014; as suggested by Reviewer #2 on an issue that is similar in spirit).

The corresponding experiments show-case the advantages and disadvantages of integrating information across cells more clearly. These are discussed in more detail in response to major comment 3 of Reviewer #2, in the results of the main manuscript and in the newly added somatic recall analysis in the Supplement (Supplementary Section S 2.5, changes highlighted main manuscript, page 43, lines 649-680).

> Calling germline mutations in single-cells therefore does not seem very useful, it is precisely the somatic mutations which inform the cell lineage trees and are important for understanding the evolution of tumors for example. However, this is the setting which the different tools are benchmarked.

We would like to point out the particular advantages of the datasets we chose in the first place.

Unlike other datasets, our datasets *decidedly address the issues relating to the benchmarking of single cell variant callers.*

Use of our datasets follows the reasoning that any single cell variant caller should be able to perform well in the most basic exercise of calling variants in single cells, regardless whether they are germline or somatic variants---because in the whole genome amplified single cell sequencing data, both germline and somatic variants will look the same, as they start with the same number of genome copies and are affected by the same errors and biases in amplification.

Thus any trust we establish in the calling of germline variants will also extend to somatic variants (give or take sufficient bulk coverage in the case of ProSolo). From that perspective, both our own dataset and that from Dong et al. (2017) are of very high value, as they support the generation of a great number of strongly reliable ground truth variants from data generated from separate sequencing experiments. The great number of reliable ground truth variants implies much enhanced reliability of the assessment of callers, for obvious statistical reasons.

The Wang et al. (2014) data nicely complements these datasets by providing more cells and, most importantly, a set of somatic mutations that were validated with a very high precision targeted sequencing method deployed at great sequencing depth. But looking at the methods section for this latter aspect nicely demonstrates the great effort that validating 467 somatic variants takes, while still not allowing to evaluate the precision of variation calls.

At the same time, such real datasets provide a much more useful picture than simulated data, as simulating data will always miss addressing some important data properties and often introduces circularity with a model's assumptions.

And finally, even in a lineage reconstruction context, germline variants and even sites that are homozygous reference in all cells can be useful information: to mitigate ascertainment bias in maximum likelihood approaches (see e.g. <https://www.biorxiv.org/content/10.1101/186478v1>). Thus, when aiming at a stepwise lineage reconstruction that first calls variants with the highest precision possible across all sites with data and then constructs lineages from the genotype probabilities at these sites, such invariant sites can be of value.

In summary, benchmarking is the major argument for our choice to work with data from fewer cells, *but a much larger, and soundly established number of ground truth variants*. At the same time, germline variants and invariant sites can nevertheless be of value for downstream analyses---even for lineage reconstruction when using maximum likelihood approaches.

> Here there are several issues with the use of SCIPhi. First this tool returns the posterior probability of mutational presence, which could be used to generate a full ROC curve. Second the runtimes seem excessive. For the case where there are two cells, there is only one possible tree, so the inference only needs to learn the parameters of the beta-binomial models (this could actually be done numerically very quickly), so the number of iterations would not need to be high (and not need to include any tree moves). The runtime grows with the number of mutations included, which is why germline mutations (from the bulk) are normally filtered out to focus on the informative data for tree lineage building.

Thanks, we appreciate the input on how to use SCIPhi, which goes beyond the documentation we have found. Accordingly, the availability of posterior probabilities returned by SCIPhi is not mentioned in any documentation we are aware of. We have now dug into SCIPhi's source code to trace the location of these posterior probabilities and have applied ProSolo's false discovery

rate control algorithm (reflecting an approved standard procedure) on them (changes highlighted supplement, page 33, lines 526-535). At any rate, we found that a sufficiently reliable FDR control is not possible with SCIPhI (see newly included results for SCIPhI in Supplementary Figure S 13, and the lack of variation in precision and recall across different values of false discovery rate thresholds in Figure 3).

Regarding runtimes, we ran SCIPhI with default parameters. There is no documentation or other guidance on good parameter choices that will give both accurate results and allow for more reasonable runtimes. In the added runs on the Wang et al. (2014) data, we reduced iterations to decrease runtime and were lucky that estimations converged on time, as a restart attempt on the whole genome dataset had previously failed without any clear remedy. Had the tool failed to converge, we would have had to rerun with more iterations, adding further weeks of wall clock time. Also, for independent corroboration of such long runtimes, see the scVILP manuscript mentioned by Reviewer #2 in their major comment 5 (e.g. Figure 4 with a runtime of around 5 hours on a simulated dataset with only 1000 mutations across 100 cells).

In particular the last point of the Reviewer seems to be critical: the quality of the trees inferred should improve on growing numbers of mutations, and as mentioned above, maximum likelihood approaches of phylogenetic reconstruction would also benefit from including clearly called invariant sites. Also, any filtering of germline variants, when done from a bulk sample, will necessarily also include some higher-AF somatic variants. Such ad-hoc filtering has thus great potential to remove variant sites that may be critical for determining lineages accurately.

Finally, runtimes will also increase when adding more cells. Note that the space of possible tree topologies grows exponentially in the number of cells, which reflects a well-known, notorious issue in tree-guided estimation.

This points out why there are clear limits to tree-guided variant calling in practice, and why it is a valuable consideration to separate the tracing of lineages from the primary calling of variants, as we now point out more clearly in the discussion of the main manuscript.

Therefore, for maintaining runtimes that scale linearly, and not exponentially, in terms of coverage and numbers of cells, we argue to do the following: first call as many variants as possible *at utmost accuracy*, and only when provided with *large amounts of high quality calls*, try to trace the lineages of cells. Following the results in this manuscript, we are ready to argue that we have some relevant advantages: we are able to deal with substantially larger numbers of mutations, we call these mutations at greater accuracy, and we have fully flexible and interpretable FDR control (which SCIPhI does not have). We are also able to systematically identify dropout and amplification errors. Overall, we would be ready to argue that ProSolo has non-negligible advantages for single cell analysis practitioners.

> *In terms of methodological advance, the beta-binomial model is (naturally) similar to other approaches from Lodato et al onwards, with novelty here a coverage-dependent drop-out term,*

and bulk contribution to the likelihood. The claim of being the first tool to offer FDR control is based on ProSolo returning posterior probabilities, but these are generated by others methods (SCIPhi for example).

We are now presenting results on the FDR control that SCIPhi can establish through the posterior probabilities it provides. In short, SCIPhi's FDR control does not work to a desirable degree, because for most reasonable levels of control (such as 0.01, 0.05, 0.1, 0.2, 0.3), SCIPhi's output hardly changes (changes highlighted supplement, page 47, lines 689-691). A closer look reveals that SCIPhi's posterior probabilities do not follow a sufficiently variable scale. Variants come in larger clusters, all of which adhere to identical probabilities. Moreover, the FDR control level desired does hardly agree with the precision achieved in reality, in all cases. With all due respect, we believe that it is fair to say that, at the very least, SCIPhi's FDR control is not truly operable in practice, thus not able to compete with ProSolo's FDR control. In summary, ProSolo remains the only tool whose probabilities allow for a fully functional FDR control.

With regard to the assessment of the novelties listed above, we agree with the Reviewer in that we are not the first ones to make use of the beta-binomial suggested by Lodato et al. (2015). However, we are the first ones to integrate the corresponding statistics into a statistical model---bringing to bear a mechanistic understanding of the whole genome amplification process---and accounting for all relevant errors and biases in a site-specific manner. From this model, various relevant probabilities (such as allele dropout or amplification errors) can be soundly derived, because we are the first ones to have a computationally efficient scheme for their computation.

This last point is likely the explanation for why our probabilities are superior in comparison with the probabilities inferred by other methods, and why we are the only ones to provide sound FDR control. Note that FDR control in variant calling from sequencing data has been traditionally a notoriously stubborn issue.

> In the methods, the parameters defining the distributions should be described (and not relegated to the supplement). Here the authors seem to use fixed values, but they should be learned from the data as part of the mutation calling (and with a prior to obtain posterior mutation calls).

Alongside the description in the main manuscript of the general parameters and their role in the defining distributions, we have now added the table with the respective values as Table 1 in the main manuscript (and removed it from the supplement; changes highlighted main manuscript, page 9).

We agree that learning these parameters from a dataset at hand is an interesting idea to pursue further, likely leading to further improvements. We discuss this in the discussion section of the main manuscript (changes highlighted main manuscript, page 26, liens 538-541).

We would like to argue further that, regardless of individual adjustment of parameters, ProSolo presents results that establish advantages over prior work, likely of great value for a practitioner already at this point (the combination of accuracy and speed, together with the first fully functional FDR control).

> To summarize, while the advances described by Laehnemann and colleagues for improved read count models for individual cells are welcome, to have a wider impact the method should include parameter estimation as well as sharing information across cells according to their phylogenetic relationships.

We agree that these are sensible suggestions. We nevertheless would like to point out again that

- ProSolo outperforms the state of the art already without these extensions, also on the dataset suggested by the Reviewer #2. Where performance of ProSolo is still lacking on these datasets, we expect a higher coverage bulk sample to further improve somatic recall. And we would like to point out that these same analyses also show false-positive imputations of somatic mutations by SCIPhI, suggesting that integrating information across cells requires considerable care.
- We decidedly focus to *not* combine variant calling with sharing information across cells. We do this based on the insight that separating the primary analysis of variants (leading to calls of utmost quality) and the secondary analysis of cell lineages (then based on variant calls of utmost quality) can have decisive advantages over combining the two kinds of analyses. Note that also in other genetics affine settings, the calling and the phasing of variants are best performed separately, while combining the tasks may lead to substantial challenges. Combining the tasks also leads to runtime explosions, because of the large space of possible phylogenies (see SCIPhI).
- Note that the probabilities that ProSolo provides do not only enable sound FDR control, but we also expect them to soundly support more accurate phylogenetic lineage reconstruction, as well. With probabilities that accurately capture the uncertainties of single cell sequencing data, phylogenetic reconstruction approaches can take the utmost care when integrating information across cells.

Here, we thus argue that further improvements on the primary issues are still desirable, given the statistically complex situation in the calling of variants in single cells, but that we have achieved quite a bit in that respect.

REVIEWER COMMENTS

Reviewer #1 (Remarks to the Author):

I would like to thank the authors for answering all my questions, and addressing my concerns regarding coverage depth of the matched bulk sample by performing additional experiments and data analysis, and adding a separate section to discuss their results. According to the authors' analysis, a minimum coverage depth of 4X should be enough to make the program function, and of course, much higher coverage depth will further improve its performance. This is understandable and most bulk sequencing data should have more than sufficient coverage depth. What troubled me is that, when benchmarking ProSolo using a real tumor dataset from Wang et al. 2014, the authors found that "the recall of ProSolo for somatic variants is lower than that for most other tools on the heterozygous alternative genotypes, especially for subclonal somatic mutations" (page 12 of the response to peer reviews). Unfortunately, one of the main purposes for the majority of single-cell DNA deep sequencing projects is to trace subclonal evolution. I suspect the static model of ProSolo depends on identifying the alternative allele using the matched bulk sequencing data. Bulk data should enhance our confidence in a mutation identified in single-cell when that same allele is also identified in bulk data, but should not make us less confident in a single-cell mutation if that same allele is not identified in bulk data. I also agree with Reviewer #3 that it would be great if ProSolo can incorporate sharing information across cells according to phylogenetic tracing. Even though this idea is already applied by SCIPHI hence not a novel idea anymore.

Reviewer #3 (Remarks to the Author):

I thank the authors for their careful responses to the previous comments, and as stated before their advances are a clear improvement on current methods. There remains however a slight contradiction in benchmarking on germline mutations and extrapolating to somatic mutations: it is not so obvious that "Thus any trust we establish in the calling of germline variants will also extend to somatic variants" given that the bulk data informs the single cell calls in ProSolo, and that calling germline mutations is obviously much easier than lower frequency ones. In this respect the addition of the Wang dataset and the response to referee 2's points help elucidate the advantages and disadvantages of sharing information across cells/using bulk data.

More minor points, including extra germline mutations help learning parameters but do not directly help with the tree structure inference since their contribution is the same for all trees and can (and should be) computed separately from the tree search. The number of trees grows super-exponentially (factorially) rather than exponentially. Tree searches/samplers are approximate algorithms that have a complexity that grows polynomially (often with a reasonably high power and without guarantees of convergence). Table 1 has far too many decimal places.

“ProSolo: Accurate and Scalable Variant Calling from Single Cell DNA Sequencing Data”

Second revision, point-by-point response

General Points to All Authors and the Editor

We sincerely thank all three reviewers for their constructive comments. Their input has helped to further improve the manuscript greatly. For instance, the detailed consideration of our assumption of a maximum of two cell populations in the bulk regarding a particular genomic site has led to a deeper appreciation of the strengths and limitations of our model.

All page and line numbers in our response below refer to the PDF documents with the “Changes_Highlighted_2nd-revision_” prefix, which will probably be added to the end of this response letter by the submission system. Further, all text changes we made are specified below as displayed indented and italicized, with added text in blue and removed text in red and with a strikethrough.

Further, we have implemented one change that is not in direct response to any of the reviewer comments and we thus include it in this general section: In re-examining the somatic recall results of the Wang et al. single nucleus exome sequencing data to answer the reviewers’ requests, we found that the alternative allele frequency from targeted duplex sequencing of some of the clonal somatic variants does not correspond to the genotype reported by Wang et al. (see Figure S 6 in the supplement). We have reclassified those for our ground truth generation and describe this in the supplement (page 31, lines 496ff):

As they provide a duplex sequencing based alternative allele frequency, we examined the alternative allele frequency distribution of the non-synonymous clonal variants, stratified by the specified zygosity (Figure S 6). This clearly shows two alternative allele frequency peaks, one at around 0.4 and the other at roughly double the alternative allele frequency around 0.7. These two peaks should correspond to heterozygous (0.4) and homozygous alternative variants (0.7), respectively. As these are supposed to be clonal variants, the remaining fraction of 0.3 beyond the homozygous alternative variants peak (from 0.7 to 1) should be contamination with normal cells (without these somatic variants). This indicates a duplex sample tumor purity of around 0.7. Unexpectedly, we also see the 0.7 peak in variants that Wang et al. [2014] classified as heterozygous and the 0.4 peak in variants that Wang et al. [2014] classified as homozygous alternative (Figure S 6). It is unclear from Wang et al. [2014] and its supplementary material how the original variant classifications were generated, we assume that this is based on the genotypes obtained for the single cells. But as the single cells were genotyped with

standard variant calling software (GATK UnifiedGenotyper), they very likely suffer from considerable allelic dropout and we thus reclassified the clonal variants' genotypes based on the duplex sequencing. If the alternative allele frequency determined by the ultra-deep duplex sequencing was below 0.6, we classified the variant as heterozygous, and as homozygous alternative if this frequency was above (or equal to) 0.6 (Figure S 6).

However, this does not change the analysis results for the calling of the presence of an alternative allele, because it only affects whether the respective variant allele is present on one or both genome copies. Also, this does not change any of the conclusions about somatic recall, as it only means that about 3 dozen variants move from the clonal heterozygous ("0/1") to the clonal homozygous alternative ("1/1") panel of Figure S 13 or vice versa.

Reviewer #1 (Remarks to the Author):

We thank the reviewer for another round of constructive review, including the consideration of the comments made by the other reviewers. They have pointed us to further clarifications and important future work.

All page and line numbers in our response below refer to the PDF documents with the "Changes_Highlighted_2nd-revision_" prefix, which will probably be added to the end of this response letter by the submission system. Further, all text changes we made are specified below as displayed indented and italicized, with added text in blue and removed text in red and with a strikethrough.

I would like to thank the authors for answering all my questions, and addressing my concerns regarding coverage depth of the matched bulk sample by performing additional experiments and data analysis, and adding a separate section to discuss their results. According to the authors' analysis, a minimum coverage depth of 4X should be enough to make the program function, and of course, much higher coverage depth will further improve its performance. This is understandable and most bulk sequencing data should have more than sufficient coverage depth. What troubled me is that, when benchmarking ProSolo using a real tumor dataset from Wang et al. 2014, the authors found that "the recall of ProSolo for somatic variants is lower than that for most other tools on the heterozygous alternative genotypes, especially for subclonal somatic mutations" (page 12 of the response to peer reviews). Unfortunately, one of the main purposes for the majority of single-cell DNA deep sequencing projects is to trace subclonal evolution. I suspect the static model of ProSolo depends on identifying the alternative allele using the matched bulk sequencing data. Bulk data should enhance our confidence in a mutation identified in single-cell when that same allele is also identified in bulk data, but should not make us less confident in a single-cell mutation if that same allele is not identified in bulk data.

We agree with the Reviewer that ProSolo is conservative insofar as its confidence in calling variants will significantly increase when bulk sequencing supports the calling. At the same time,

by modelling the bulk sample jointly, ProSolo's statistical model can ensure that variants, when called, are indeed variants and not amplification errors.

We apologize for not having communicated that more clearly and have thus added a respective sentence in the Results section (page 9, lines 181f):

At the same time, the absence of a single cell candidate mutation in a bulk sample (with sufficient sequencing depth) increases the probability of an amplification error.

In the case of the Wang et al. dataset, a deeper sequencing of the bulk would easily raise ProSolo's (somatic) recall, as this would provide data to much better distinguish between artifacts and true phenomena. Even for subclonal heterozygous variants, the probability that the bulk supports these variants continuously increases with increasing sequencing depth. With ProSolo's model, this does not simply mean that ProSolo can either call a variant or not, but is a more fine-grained question of how strong the differences between the probabilities of the various defined events will come out. More bulk coverage just means more data and thus more confident event classifications, even for subclonal variation.

One thing the reviewer has made us realize---for which we are grateful---is that the current model setup lacks a prior for somatic mutations in the bulk sample. Note that by its (Bayesian) setup, our model makes it possible to integrate such a prior. Based on the somatic mutation rate, the prior would provide an important baseline probability where bulk coverage is too low for reads to confidently sample an alternative allele. It would provide non-zero likelihoods to sites where no alternative allele reads are present in the bulk, but the impact of the prior would wane with increasing coverage in the bulk sample (and thus with increasing confidence that a variant genuinely is not present there if no read indicates it). A newer version of the computational library that ProSolo makes use of already implements such a prior. It is therefore, also in practice, interesting future work to put it to use for ProSolo. We have added this point to the discussion (page 26, lines 541ff):

In addition, the current model prevents the calling of subclonal somatic mutations with a bulk sample that has insufficient coverage to sample the respective allele, which can easily be remedied by sequencing the bulk sample to a greater depth. In addition, including a prior for the somatic mutation rate in the bulk sample should further improve recall. Newer versions of the Varlociraptor library³¹ already allow using such a prior, so this is immediate future work. When bulk coverage is low and an alternative allele is not sampled by any read, the prior will result in non-zero alternative allele frequencies in the bulk being assigned non-zero likelihoods, which is the desired behaviour for this constellation. When bulk coverage is high, the increased amount of data will progressively overrule the prior.

In addition, adjusting the whole genome amplification distribution used for homozygous sites should further improve the recall of subclonal heterozygous alternative somatic mutations when they are subject to strong amplification bias against the alternative allele in a single cell. This is discussed as future work in the Supplementary Section S 1.2.2 ("Outlook: more realistic beta-binomial distributions") and mentioned in the Discussion.

I also agree with Reviewer #3 that it would be great if ProSolo can incorporate sharing information across cells according to phylogenetic tracing. Even though this idea is already applied by SCIPhI hence not a novel idea anymore.

While SCIPhI nicely demonstrates the benefits of sharing information across cells, we would nevertheless like to reiterate our advocacy for our approach in the name of scalability, because involving tree construction into the variant calling process can be computationally extremely demanding, and quickly reaches its limits, both in theory (the space of possible trees is super-exponential in the number of single cells), and in practice. As long as information is not shared across single cells during variant calling, variant calling can be done in parallel across cells. In addition to profiting from parallelizing the calling step, phylogenetic inference can be addressed subsequently, using heavily specialized and very fast software (like RAXML, see also below), which in terms of runtime outperforms SCIPhI by large amounts.

In a bit more detail, SCIPhI uses one of the classical approaches to make this exploration tractable for larger numbers of cells, a Markov chain Monte Carlo method. However, as can be seen from the runtimes we report, even when using such approximations, SCIPhI's runtimes are considerably higher (counted in weeks) than for other approaches (at most 2-3 days).

Thus, as we are able to improve or at least be on a par with SCIPhI in terms of calling performance, while having runtimes orders of magnitude smaller, we would advocate for investing into phylogenetic inference only *after* variants have been called, instead of simultaneously. As the variant calling results (the posterior probabilities computed) comprehensively and accurately capture uncertainties of the data, this reduced representation of the data computed by ProSolo can subsequently be used for sharing data across cells in downstream analyses in a statistically sound way.

Note, finally, that there are more efficient probabilistic phylogenetic inference approaches. RAXML-NG (<https://doi.org/10.1093/bioinformatics/btz305>) is such an approach and can make use of ProSolo's genotype-specific posterior probabilities and is thus able to carry over the uncertainty of the data that ProSolo captures into its phylogenetic inference.

In summary, we advocate for investing in both the calling of variants and the reconstruction of trees separately. We are planning to explore the construction of pipelines to encompass both steps in future work.

Reviewer #2 (Remarks to the Author):

We thank the reviewer for once again considering our model and especially the newly added information in great detail. This has led to a deeper appreciation of the strengths and limitations of our model.

All page and line numbers in our response below refer to the PDF documents with the “Changes_Highlighted_2nd-revision_” prefix, which will probably be added to the end of this response letter by the submission system. Further, all text changes we made are specified below as displayed indented and italicized, with added text in blue and removed text in red and with a strikethrough.

In the revised version, the authors have added the results of ProSolo on an exome-sequencing dataset with larger number of (32) cells and a downsampling experiment. These new results certainly improve the manuscript. However, some issues still remain, and I found new issues in the presented results of the new experiments which need to be addressed.

- 1. In order to address my major comment 1, the authors have updated the description figure 1D. Now the authors have replaced the of θ_s and θ_b with $\tilde{\theta}_s$ and $\tilde{\theta}_b$ and these new parameters are meant to refer to the likelihood density estimates across the respective allelic frequencies as raised by ProSolo. However, the manuscript does not discuss how these density estimates are actually related to the true alternate allele frequencies or the observed alternate allele frequencies. As per my understanding, these new parameters reflect the assumptions made by ProSolo’s model assuming no copy number alteration. However, it is still not clear how $\tilde{\theta}_b$ is related to the observed variant allele frequency in bulk sequencing data. Also, how it is estimated is not clear from the manuscript.*

The likelihood densities plot the likelihood of alternative allele frequencies (single cell: θ_s ; bulk: θ_b) given the sequencing data for a particular genomic locus. Computation of these densities is considerably more involved for θ_s because of the amplification step, which requires computing an integral as an intermediate step. In this computation, θ_s is inferred from ρ_s , where ρ_s reflects the alternative allele frequency of amplified data. Because the amplification tends to heavily distort the proportion of alleles, ρ_s usually differs from θ_s ; in particular ρ_s can be anything between 0 and 1, while θ_s can only assume the values 0, 0.5, or 1 (see equation 2 in the main manuscript and equation S24 in the supplement).

Estimating θ_b , the alternative allele frequency for the bulk, is substantially less involved, because the amplification step is missing. See equation 3 in the main text and equation S 21 in the Supplement. See also Reference 31 in the main text for further details on computing likelihood density estimates for variants in bulk sequencing data.

From a general perspective, θ_s and θ_b are true, but hidden values, because they cannot be computed directly from the data given, but only estimated by means of a reasonable algorithm. [An analogy is the true mean μ of a Gaussian distribution from which one has sampled data: the average $\tilde{\mu}$ across the data is an estimate for μ given the data.]

For distinguishing estimates as provided by us, from the true (but unknown) θ_s and θ_b , we have introduced notations $\tilde{\theta}_s$ and $\tilde{\theta}_b$. Of course, the only thing we can work with in practice are the (likelihood density) estimates computed by us, but not the hidden true values. Therefore, definitions of events refer to the estimates, integrating over the corresponding alternative allele frequencies. As the quality of our estimates is witnessed by the results we report, namely that they are excellent as long as the data is sufficiently certain, we are confident that the model works very well for providing those estimates and interpreting them jointly for the single cell and bulk sample.

We have added and adjusted the text to make the distinction between the different notations clearer:

- main manuscript page 5, line 122:
 - ~~theoretical~~ true (but usually unknown)
 - main manuscript page 5, lines 129f:
 - *The goal is to estimate the likelihood density across the three possible underlying allele frequencies in the single cell (we denote $\tilde{\theta}_s$ as the density estimate across all $\theta_s \in \{0.0, 0.5, 1.0\}$).*
 - main manuscript page 9, lines 190ff:
 - *Further, as for the single cell sample, when referring to the likelihood density estimates across all possible allele frequencies (as opposed to a particular allele frequency), we denote this with $\tilde{\theta}_b$.*
 - supplement page 2, lines 40ff [including footnote 2]:
 - *It is our eventual goal to work back from bulk and single cell data to derive likelihoods for all possible values of $\theta_b(x)$ (denoted as $\tilde{\theta}_b(x)^2$) and $\theta_s(x)$ (denoted as $\tilde{\theta}_s(x)$), respectively, allowing us to place likelihoods on possible underlying genotypes and events like allele dropouts in single cells. For a single diploid cell s , the ~~possible alternative nucleotide frequencies are true~~ alternative nucleotide frequency can be:*
 - *²We use θ notation for the true underlying alternate allele frequency at a site, and $\tilde{\theta}$ notation when referring to the likelihood density estimates derived for all possible underlying alternate allele frequencies. The latter notation is thus mostly used for the definition of single cell events in section S 1.5.1.*
 - supplement page 16, line 248:
 - inserted $\tilde{\theta}_s$ and $\tilde{\theta}_b$ at the respective spots
2. *The authors have explicitly mentioned their assumption on bulk sequencing data. Now, they consider that the bulk reflects a mixture of at most 2 cell subpopulations with regard to any genomic site. However, they have not provided explicit rationale for these assumptions. Specifically, for $\tilde{\theta}_b \in [0.5, 1)$, this assumption may get violated, e.g.,*

heterozygous population with cellular frequency 0.6, homozygous reference population with frequency 0.1 and homozygous alternate population with frequency 0.3 can give rise $\tilde{\theta}_b = 0.6$. The authors should explore the performance of ProSolo when such to violations are present in the dataset. Clearly, simulation experiments need to be designed for such purposes. Without experimental support for such assumption, it is not clear how reasonable these assumptions are given that they are restrictive. For example, in cancer subsequent deletion of the reference allele in a heterozygous genotype can give rise to a situation where 3 cell subpopulations (homozygous ref and alt, heterozygous) can be present in a mixture.

We thank the reviewer for pointing us to cases where more than two cell subpopulations were observed, and for providing further references when we asked back for clarification of the scope of the question. We have used the provided references as the basis for a thorough literature search on cancer mutational events that violate the infinite sites assumption. As we look at every single cell separately, we do not make the assumption of no parallel evolution (the same mutation happening through distinct events in different cell divisions) in our setup, and can thus disregard this aspect of the infinite sites assumption for our considerations. Instead, we focused on the consequences of a violation of the assumption of two cell populations in the bulk for ProSolo's calling performance.

Cases where the infinite sites assumption is violated exist, in amounts that are above the general expectations (by the assumption). We agree with the Reviewer that it is important to not neglect such cases, but examine them. At the same time, variants yielding more than two subpopulations can be considered "corner cases", because they arise in quantities that are by orders of magnitudes smaller than variants inducing two subpopulations.

But even when dealing with three subpopulations, ProSolo still tends to make the correct calls in the great majority of cases and cells, without having to explicitly model it. To exemplify this argument, we consider the example given by the reviewer. According to its model, which will calculate a very high likelihood density at $\tilde{\theta}_b = 0.6$, ProSolo assumes two subpopulations: cells that are heterozygous and cells that are homozygous for the alternative allele. As a consequence, ProSolo still classifies 90% of the cells in the example correctly (those with a heterozygous or homozygous alternative genotype). Only for 10% of the cells in the example (the ones that are homozygous for reference), ProSolo will assign the highest likelihood to 'Allele Dropout to Reference' instead of 'Homozygous for Reference'. Note that, although being incorrect in the interpretation, this is still correct insofar as ProSolo correctly realizes the absence of alternative alleles in the single cell (but misinterprets it because homozygous for reference is not part of the scenario for this bulk alternative allele frequency). However, we have not been able to find an example that would correspond to such allelic frequencies in the literature.

As for evidence in the literature of multiple mutational hits at a particular genomic site, these all seem to arise by the same mechanism (which is also mentioned by the reviewer in their

question): (i) a point mutation introduces a mutant allele, creating a heterozygous genotype alongside the original homozygous genotype; (ii) a loss-of-heterozygosity event deletes one of the alleles, generating a new subpopulation. However, only if the original homozygous allele is lost, a distinguishable third subpopulation is generated. We found one such instance in the literature, which to our knowledge is the only published real case where the two-population assumption in the bulk sample would lead to a ProSolo misclassification: the homozygous alternative site would be called as heterozygous, as the highest likelihood would be assigned to the “ADO to alt” event (Figure 1D). As this potential misclassification directly follows from our model setup—in the sense that errors can be immediately predicted and classified—we think that simulations would not provide any insight that goes beyond the analyses provided here. Note at last, that this mistake does not affect the calling (the presence of an alternative allele is correctly called), but only the genotyping.

In the main text, we now mention the possible issue and point out the possible consequences in the discussion (page 27, lines 562ff):

Since While no standardized software is available, it seems to be possible to obtain copy number profiles from single cells 52, 53 . Extending our model to make use of such profiles, it should also be possible to salvage cancer-related variant cases that our current model setup does not capture (Supplement, Section S 1.5.1). And since [...]

This addition also points to our belief that copy number aware approaches can salvage such ‘corner cases’, as soon as easily usable standardized approaches for copy number determination from single cell DNA sequencing data are available.

Finally, and most importantly, we also provide the above discussion of possible violations of our two subpopulation assumption in the bulk sample in more detail and with all the relevant citations in the supplement and point to it from the Results section of the main manuscript (page 10, lines 202f):

For these definitions, we always assume that—regarding a particular genomic site—the bulk cell population can only consist of a maximum of two sub-populations that are exactly one mutated allele apart from each other (Figure 1D, lower panel; for discussion on the validity of this assumption, see Supplementary Section S 1.5.1).

The full new text in Supplementary Section 1.5.1 starts on page 18, lines 268ff:

To evaluate whether this assumption of only two sub-populations in the bulk is safe to make, we can examine all the possible ways in which this assumption may be violated. This requires that we have a sequence of two mutational events affecting the same genomic site, each generating a new sub-population. To know which types of mutational event sequences to consider, we note that the same point mutation has been observed to occur independently in different single cells from the same individual [Kuipers et al., 2017], possibly due to selection for an advantageous effect of that mutation. However, the net effect of such parallel evolution still restricts the cell bulk to two sub-populations with distinct genotypes¹². At the same time, we are not aware of cases where different

cells from the same population acquire different point mutations at the exact same locus, and expect this to be highly unlikely. Instead, a third sub-population will usually be generated by a loss of heterozygosity event on a larger scale, covering a mutated site [Kuipers et al., 2017, Satas et al., 2020]. Thus, if we start with a homozygous reference bulk, a first event would introduce an alternative allele mutation and a second event in that sub-population would lead to the loss of either the reference allele or the alternative allele. Losing the recently generated alternative allele leads to an allele frequency in that sub-population that is similar to the homozygous reference genotype (it only has coverage of the reference allele), and this sub-population is thus indistinguishable from the original homozygous reference population. ProSolo will correctly identify any single cells with the loss event as homozygous reference (Figure S 2). However, losing the reference allele leads to a sub-population that only has coverage of the alternative allele and is thus similar to a homozygous alternative genotype and clearly distinguishable from the other two genotypes. While this type of event was not observed in a systematic survey of multiple mutational hits per site [Kuipers et al., 2017], it has coincidentally been observed in a recent study that explicitly modeled copy number alterations in single cells for their phylogenetic reconstruction of cell lineages (mutation LINGO2:2 in Satas et al. [2020]). Thus, we expect this type of event to be rare, but possible. In this case, as the alternative allele was only introduced in a sub-population and the loss of the reference allele is only present in an even smaller sub-population of that, the bulk frequency of this alternative allele will be well below 0.5. Here, the ProSolo model assumes the homozygous alternative genotype of this sub-population impossible and instead calls cells from that sub-population as heterozygous (Figure S 2). However, this only concerns the genotype, while the presence of the mutation will be identified correctly. As MonoVar is the only other variant calling software that can even genotype beyond calling the presence or absence of a mutation, this could only be considered a drawback compared to that tool. As germline homozygous alternative sites are very rare compared to germline homozygous reference sites (due to the reference genome being biased towards more common alleles), we expect the symmetric case¹³ to be much less likely to be observed. Finally, starting from a heterozygous germline genotype in the bulk will be more likely than starting from a homozygous alternative genotype, but a lot less likely than starting from a homozygous reference genotype. From this starting point, three sub-populations regarding the same alternative allele can only be generated by two events in opposite directions in distinct cells from that population: one point mutation or loss event towards the reference and one point mutation or loss event towards the alternative allele, with the loss events much more likely to affect any site than the point mutations. Whether the resulting bulk alternative allele frequency then ends up above or below 0.5, depends on which event happens earlier and/or grows faster within the population. As a result, the homozygous genotype (reference or alternative) with the lower frequency will erroneously be considered a heterozygous single cell genotype, but ProSolo will correctly call the more frequent (and thus probably more relevant) homozygous genotype. However, we are not aware of any such events being reported in the literature and expect them to be vary rare. We conclude that the assumption is reasonable to make and we only expect it to lead to the misclassification of homozygous

alternative single cell sites as heterozygous when they have been generated by the introduction of an alternative allele via a point mutation, followed by an allelic loss of the reference allele in some daughter cells. We also discuss possible respective cases in the context of the tumor single cells from the data of Wang et al. [2014] (Supplement Section S 2.5). It should be possible to mitigate such cases with further extensions of the model to account for copy number changes. As in Satas et al. [2020], this would require separately obtaining copy number profiles that are then provided as input to ProSolo, but we do not yet see a standardized way of obtaining such profiles from single cell DNA-sequencing data. So far, such copy number profiles seem to have usually been obtained with custom scripts tailored to specific datasets (for example Baslan et al. [2012], Satas et al. [2020], Kuipers et al. [2020]), but when more generalized approaches for obtaining such profiles become available, they could then be used to adjust the exact model setup for each site based on the local copy number profile. Newer versions of varlociraptor, whose variant calling functionality ProSolo uses in the background, already allow for a chromosome-specific specification of copy numbers¹⁴. Thus, it should be possible to extend this to site-specific copy numbers in the future and to extend the ProSolo model to arbitrary ploidies.

- ¹² As we are looking at different single cells separately, this also means that we do not have to worry about this kind of violation of the infinite sites assumption. Such parallel evolution only becomes problematic for phylogenetic reconstruction approaches that assume infinite sites, which would assume that each mutation only happens once in the tree.
- ¹³ A mutation towards the reference with a subsequent complete loss of the originally homozygous alternative allele.
- ¹⁴ See the "ploidy" definition in this documentation section:
<https://varlociraptor.github.io/docs/calling/#configuring-the-joint-prior-distribution>

3. In my previous review, I particularly mentioned that imputation based on bulk cell population will be problematic for somatic mutation sites. The authors have argued that their model performs better even without using imputation. However, the previous subsection titled "Biologically relevant imputation based on the bulk sample" remains as it is. Imputation based on bulk sample is only meaningful for clonal mutations. While the authors have added a paragraph in the Discussion section, the aforementioned subsection in Results need to be more precise in its description and should clearly mention in which cases the bulk provides meaningful imputation.

We apologize for this oversight and have amended the respective Results section of the main manuscript (page 13, lines 254ff):

*However, while this is a biologically meaningful way of imputation at the vast majority of genomic sites, it should be noted that this imputation will **usually** favor germline genotypes over any existing (lower frequency) somatic genotypes at a site, **unless such a somatic genotype is present in a majority of cells**. Thus, such an imputation carries the potential to introduce erroneous calls (especially when looking at subclonal somatic*

mutations), and we recommend to instead use downstream tools that can accommodate for missing data and data uncertainty wherever they are available.

In addition, we also added to the Abstract the caveat that imputation should only be used if downstream tools require it (main manuscript, page 2, lines 31ff):

[...] and supports imputation of genotypes in single cells at insufficiently covered ~~polymorphic sites~~ sites, when downstream tools cannot handle missing data. As a consequence, ProSolo achieves [...]

- 4. The analysis of Wang et al. dataset shows that SCIPhI performed better than ProSolo for the normal cells. While the authors compared the F1 score of different algorithms for the other 2 datasets, they have not presented so for this new dataset. They should add F1 score comparison for Wang et al. dataset. For tumor cells also, SCIPhI seems to perform better in terms of recall. While ProSolo achieves above 0.99 precision for the tumor cells, the recall is very low for such cases. Comparison of F1 score will give a clearer picture.*

While we understand and agree with the concern, we did not include F1 score plots for the new Wang et al. dataset purposefully. The reason is that F1 scores provide a false sense of comparability across tools, and are rarely informative in practice: note that for determining F1 scores, one needs to determine the corresponding parameter (which is called ‘threshold used’ in Figure 3) that leads to optimal F1 for each of the tools individually, but ProSolo is the only tool where that threshold has a meaningful interpretation (the false discovery rate). As a result, the same threshold in different tools will mean something completely different, and how to determine a good threshold value in practice (i.e. in the absence of a ground truth) is only clear for ProSolo.

Thus, to showcase variation between individual cells, we opted for a direct comparison of precision and recall on individual cells for the Wang et al. dataset, but originally kept the existing F1 score plots for the previous datasets for consistency with the preprint. We have now opted for consistency across datasets and exchanged all the former F1 score plots with cell-specific precision recall plots, to showcase the (minimal) variation of results across single cells (Supplementary Figures S8 to S11). Further, we now explicitly note in all the captions of figures where “threshold used” is indicated, that these thresholds are not comparable across tools.

We believe that the results we present sufficiently clarify the situation on the Wang et al. dataset: in terms of overall performance, SCIPhI comes out first, and ProSolo second; but ProSolo is the only tool that allows one to flexibly adjust FDR control. Looking at the somatic recall in detail (Supplement Figure S13) reveals that SCIPhI tends to overimpute (as can be seen from calling subclonal variants in too many cells), while ProSolo tends to be very conservative (as can be seen from not calling subclonal variants in many cases). As a final remark, let us add that the data set maximally favors SCIPhI, because of the availability of sufficiently many single cells, but does not do a particular favor for ProSolo, because a sufficiently deeply sequenced bulk sample is not available. This explains the slight advantages of SCIPhI.

- 5. The new figure showing the results of downsampling experiment is too cluttered. I suggest separating that into 2 figures, one in which the bulk coverage is varied (threshold being fixed) and F1 score is compared across different bulk coverage, another where the threshold is varied.*

We agree that the figure looked cluttered, thanks for pointing this out. As we also mention in the response to point 4 above, the thresholds of different tools are not really comparable with one another. Thus, fixing a particular threshold value will not be a like-for-like comparison and not interpretable in a practically meaningful way. To nevertheless improve the legibility of the downsampling experiment results, we have split the figure into two separate panels, one focusing on the high-precision region and another also encompassing the lower precision results at a bulk coverage of 0X.

- 6. In discussing the results of SCIPhI on the Wang et al. dataset, the authors mentioned that SCIPhI over-imputes mutation. The reasoning they provide is that the maximum frequency for a heterozygous subclonal mutation in tumor bulk was 0.233, while SCIPhI calls nine heterozygous mutations in 12 or 15 cells. The claim of the authors here is incorrect in my understanding. The single cell samples cannot be treated as ideal, uniformly sampled data. Surely these samples are biased. While the bulk data represents the average profile, the single cells are random samples and can harbor mutations in a different frequency as compared to the bulk. In fact, rare somatic mutations that are not detected in the bulk can also be detected from single cells. Also, $VAF=0.233$ does not ensure that only 23.3% tumor cells harbored a mutation. Most often the bulk tumor samples are contaminated with normal cells and that can reduce the VAF of a heterozygous mutation. VAF is also affected by copy numbers and cellularity of the tumor clone harboring that mutation. Mere difference in the mutation frequencies does not suggest that SCIPhI's model over-imputes.*

Looking at the methods section of the original Wang et al. 2014 paper suggests that frozen tissues were dissociated and single nuclei sorted out via fluorescence activated cell sorting that only gated on the cell cycle stage of the nuclei. On tumor tissue, this procedure should include the nuclei of contaminating normal single cells. We would thus assume that the sampling of cells should be rather uniform across the section that was dissociated for this sequencing and that the bulk VAF is representative of a tumor sample including normal contaminating cells.

It is, however, possible that the bulk sequencing and the targeted duplex sequencing were performed on different distant parts of tumor tissue with entirely different subclones, as the method description does not provide any detail about the tumor regions used for the different types of sequencing. At the same time, this would have defeated the purpose of the generated data in the original study, and we thus assume that close-by regions of the tumor (and cells from the same normal sample in the case of normal cells) were targeted with all the approaches. Thus, the VAFs from the very deep targeted duplex sequencing should at least roughly correspond to the sampling of single cell nuclei.

Even though a sampling of 16 cells could oversample a particular subclone by chance, to have a mutation that was found in only 5 or 6 cells in the original study, but is found in 15 out of 16 cells here, remains very unlikely. This would require a doubling of the VAF of 0.233 (the highest among these variants) to a VAF of 0.469 (to explain a heterozygous somatic mutation in 15 out of 16 cells), and even stronger sampling deviations for the other variants with lower targeted duplex sequencing VAFs.

Reviewer #3 (Remarks to the Author):

We thank the reviewer for once again stressing ProSolo's clear improvements over current methods and for their detailed consideration of our previous responses.

All page and line numbers in our response below refer to the PDF documents with the "Changes_Highlighted_2nd-revision_" prefix, which will probably be added to the end of this response letter by the submission system. Further, all text changes we made are specified below as displayed indented and italicized, with added text in blue and removed text in red and with a strikethrough.

I thank the authors for their careful responses to the previous comments, and as stated before their advances are a clear improvement on current methods. There remains however a slight contradiction in benchmarking on germline mutations and extrapolating to somatic mutations: it is not so obvious that "Thus any trust we establish in the calling of germline variants will also extend to somatic variants" given that the bulk data informs the single cell calls in ProSolo, and that calling germline mutations is obviously much easier than lower frequency ones. In this respect the addition of the Wang dataset and the response to referee 2's points help elucidate the advantages and disadvantages of sharing information across cells/using bulk data.

We thank the reviewer for raising this point and agree that the addition of the Wang et al. dataset suggested by reviewer #2 was crucial in highlighting this.

As can be seen on that dataset, our above cited statement does not hold in the case where a subclonal variant has a low frequency in the cell population and bulk sequencing is not deep enough to sample it. Just to prevent misunderstandings: with our sentence from above, we were meaning to point out that in the single cells themselves, all (apart from CNV affected) variants have frequencies 0, 0.5, or 1. In that sense, calling germline variants in single cells creates trust also in calling somatic variants. This of course differs when dealing with bulk data and low frequency variants. At the same time, supported by the additional experiments on varying bulk coverage, we are confident that deeper bulk sequencing can remedy this easily.

More minor points, including extra germline mutations help learning parameters but do not directly help with the tree structure inference since their contribution is the same for all trees and can (and should be) computed separately from the tree search.

Yes, extra germline mutations will not affect the tree structure / topology, but will have an effect on estimated branch and tree lengths, and can thus influence estimates of (e.g. evolutionary) parameters based on the reconstructed tree.

The number of trees grows super-exponentially (factorially) rather than exponentially. Tree searches/samplers are approximate algorithms that have a complexity that grows polynomially (often with a reasonably high power and without guarantees of convergence).

We thank the reviewer for pointing these things out and have adjusted the respective wording in the manuscript from “exponentially” to “super-exponentially”:

- main manuscript page 20, line 382:
 - *In addition, adding more cells grows the space of possible tree topologies that SCIPhI explores ~~exponentially~~super-exponentially¹³, which will further increase its runtime.*
- main manuscript page 24, line 483:
 - *Thus, any addition of genomic coverage and cells will further increase SCIPhI's wall clock runtime, where adding cells could prove especially troubling: adding cells will ~~exponentially~~super-exponentially grow the space of possible tree topologies to explore.*

Table 1 has far too many decimal places.

As the decimal places are a reflection of the precision used in our implementation, we have now moved the table with the full precision back to the supplement for documentation purposes. In Table 1 in the main manuscript, we have reduced the number of decimal places as far as possible without losing information on the slope of α .

REVIEWERS' COMMENTS

Reviewer #1 (Remarks to the Author):

This reviewer voiced two concerns/suggestions in my last report:

a. This reviewer was concerned with ProSolo's performance with data where single-cell mutations were not identified by bulk data. The authors acknowledged this concern and addressed it by adding a prior in their algorithm to avoid over-penalizing these under-covered genomic regions in bulk data.

b. Both Reviewer #3 and this reviewer suggested that ProSolo may incorporate the phylogenetic tree inference method from SciPhi into ProSolo to further improve its performance. The authors instead proposed an alternative: using ProSolo to call mutations and developing a separate workflow to construct a single-cell phylogenetic tree. I agree with the authors' proposal since this may allow future users to interpret single-cell somatic mutations first and inject their insights before phylogenetic inference.

Overall I think the authors have addressed all my concerns.

Reviewer #2 (Remarks to the Author):

I thank the authors for their sincere responses to my comments. The authors have reasonably addressed my comments regarding the estimation of some model parameters, and possible violations of their model assumptions. I am happy with the newly added discussion on how their model would work in the event of the presence of three subpopulations. Personally, even though I feel that a simulation experiment would have further strengthened their argument, I accept their decision of not conducting any simulation experiment in this regard. At this point, I just have a minor comment

1. While I partially agree with the argument of the authors in response to my point 6 that SCIPHI may be over-imputing mutations for some genomic sites, it would be helpful if such genomic sites are highlighted and the results are contrasted with that of ProSolo. The authors mention that SCIPHI does this for 9 sites. Can the authors visualize the mutation matrix inferred by ProSolo and SCIPHI and highlight these mutations? The authors can adopt the heatmap visualization as used in the original study (Fig. 3e Wang et al. 2014). It will also be helpful to uncover the subclonal structure revealed by the called mutations from each of the methods.

Reviewer #3 (Remarks to the Author):

I thank the authors for their further clarifications and revisions, which address all concerns.

ProSolo, final (3rd) revision: point by point response to reviewers

We thank all reviewers for their comments and suggestions, and the editorial team for their dedication to well-documented methods. The only change in content in this last round is from addressing point 1 raised by Reviewer #2 (see below). All other changes found in the PDFs with changes highlighted address editorial requests. Especially in the main manuscript, this means that a lot of text appears as both red (deleted) and blue (newly added), as the section order had to be adjusted to the journal's specification. Further, please note that cross-references from the main manuscript to the supplement and vice versa are broken in the "Changes-highlighted_3rd-revision_" PDFs. We apologize for this, but the documents clearly show what has been changed. The references do work in the documents provided for production.

Reviewer #1 (Remarks to the Author):

This reviewer voiced two concerns/suggestions in my last report:

a. This reviewer was concerned with ProSolo's performance with data where single-cell mutations were not identified by bulk data. The authors acknowledged this concern and addressed it by adding a prior in their algorithm to avoid over-penalizing these under-covered genomic regions in bulk data.

b. Both Reviewer #3 and this reviewer suggested that ProSolo may incorporate the phylogenetic tree inference method from SciPhi into ProSolo to further improve its performance. The authors instead proposed an alternative: using ProSolo to call mutations and developing a separate workflow to construct a single-cell phylogenetic tree. I agree with the authors' proposal since this may allow future users to interpret single-cell somatic mutations first and inject their insights before phylogenetic inference.

Overall I think the authors have addressed all my concerns.

We thank the reviewer for the careful consideration of our revisions.

Reviewer #2 (Remarks to the Author):

I thank the authors for their sincere responses to my comments. The authors have reasonably addressed my comments regarding the estimation of some model parameters, and possible violations of their model assumptions. I am happy with the newly added discussion on how their model would work in the event of the presence of three subpopulations. Personally, even though I feel that a simulation experiment would have further strengthened their argument, I accept their decision of not conducting any simulation experiment in this regard. At this point, I just have a minor comment

1. *While I partially agree with the argument of the authors in response to my point 6 that SCIPhI may be over-imputing mutations for some genomic sites, it would be helpful if such genomic sites are highlighted and the results are contrasted with that of ProSolo. The authors mention that SCIPhI does this for 9 sites. Can the authors visualize the mutation matrix inferred by ProSolo and SCIPhI and highlight these mutations? The authors can adopt the heatmap visualization as used in the original study (Fig. 3e Wang et al. 2014). It will also be helpful to uncover the subclonal structure revealed by the called mutations from each of the methods.*

As the reviewer suggested, we now highlight the relevant mutations in the supplement. As we wanted to also include all other tools in the comparison, we chose to present the mutation frequency among the 16 single tumor cells in the form of a table. This nicely demonstrates that the tools generally agree with the cell numbers reported for each mutation by Wang et al., with a maximum of two more cells reported to bear any particular mutation. One exception is a mutation, where all tools report higher cell numbers (9-11) than the original paper (6). However, SCIPhI is the only tool to report an even higher number of cells (15) and in the other 8 mutations, SCIPhI is the only tool with strongly elevated cell counts. Thus, SCIPhI's cell counts in these cases not only contradict the original report, but also a consensus across the other tools. This further suggests that SCIPhI overimputes the mutation in these cases.

Reviewer #3 (Remarks to the Author):

I thank the authors for their further clarifications and revisions, which address all concerns.

We thank the reviewer for their consideration of our revisions.